# Learning to Discuss Strategically:
# A Case Study on *One Night Ultimate Werewolf*

**Xuanfa Jin**[*,1,3], **Ziyan Wang**[*,2], **Yali Du**[†,2], **Meng Fang**[4], **Haifeng Zhang**[†,1,3,5], **Jun Wang**[†,6]

[1]Institute of Automation, Chinese Academy of Sciences,
[2] Cooperative AI Lab, Department of Informatics, King's College London,
[3]School of Artificial Intelligence, University of Chinese Academy of Sciences,
[4]University of Liverpool, [5]Nanjing Artificial Intelligence Research of IA,
[6]AI Centre, Department of Computer Science, UCL

## Abstract

Communication is a fundamental aspect of human society, facilitating the exchange of information and beliefs among people. Despite the advancements in large language models (LLMs), recent agents built with these often neglect the control over discussion tactics, which are essential in communication scenarios and games. As a variant of the famous communication game Werewolf, *One Night Ultimate Werewolf* (ONUW) requires players to develop strategic discussion policies due to the potential role changes that increase the uncertainty and complexity of the game. In this work, we first present the existence of the Perfect Bayesian Equilibria (PBEs) in two scenarios of the ONUW game: one with discussion and one without. The results showcase that the discussion greatly changes players' utilities by affecting their beliefs, emphasizing the significance of discussion tactics. Based on the insights obtained from the analyses, we propose an RL-instructed language agent framework, where a discussion policy trained by reinforcement learning (RL) is employed to determine appropriate discussion tactics to adopt. Our experimental results on several ONUW game settings demonstrate the effectiveness and generalizability of our proposed framework. The project page of our paper: one-night-ultimate-werewolf.github.io.

## 1 Introduction

Many games such as StarCraft [1, 2], Diplomacy [3, 4] can approximate various fundamental issues in real life, and studying these games contributes to a better understanding of the functioning of our society [5]. Designing artificial intelligence (AI) agents that can play these games well has attracted a lot of attention [6, 2, 3]. Fortunately, recent large language models (LLMs) have demonstrated significant potential in constructing intelligent agents in numerous tasks [7–10] due to their impressive reasoning and emergent generalization abilities [11–13]. Moreover, LLM-based agents have achieved approaching or even surpassing human performance in games like Chess[14], Minecraft [15–17], Avalon [18–20] and Werewolf [21–23], etc.

In the game Werewolf [24], the hidden roles and uncertain discussions greatly influence the gameplay, making it challenging for players. Compared to it, the *One Night Ultimate Werewolf* (ONUW) game adds roles that can change the roles of players, and all players only have one nighttime to take action and one daytime to discuss and vote. Therefore, the key challenge of the ONUW game is to deduce

---

[*]Equal contribution. Author ordering is determined by coin flip.
[†]Correspondence to: ⟨yali.du@kcl.ac.uk⟩, ⟨haifeng.zhang@ia.ac.cn⟩, ⟨jun.wang@cs.ucl.ac.uk⟩.

38th Conference on Neural Information Processing Systems (NeurIPS 2024).

and distinguish the final roles of all players from their statements. Meanwhile, due to the sequential execution of actions at night, the information obtained by the former player might be not reliable, leading to the rise of reasoning difficulty and uncertainty. For example, Seer and Robber are two roles in the ONUW game with special abilities, and the Seer saw a player was a Werewolf at night. However, if the Robber switched that player's role later on, the information obtained by Seer becomes invalid. So players need to discuss strategically, leveraging others' statements and their own prior knowledge to guide other players to reveal their information, or conceal their identities by misleading or deceiving, particularly for Werewolves. Recent works [21, 22, 19] have attempted to construct LLM-based agents that can play communication games, but they mainly focus on how to fully utilize reasoning and generalization abilities of LLMs, while neglecting the control over strategies.

In this work, we focus on enhancing the discussion ability of LLM-based agents by leveraging the ONUW game. We formulate the ONUW game as a Multi-Phase Extensive-Form Bayesian Game and explore various discussion tactics in the game. Through analyzing a three-player ONUW game with two Werewolves and one Robber, we present the existence of the Perfect Bayesian Equilibria (PBEs) [25] in two distinct scenarios: one with discussion and one without. Players' utilities at equilibria when in the scenario with discussion highlight the significance of discussion, as they are only determined by players' beliefs, which are influenced by discussion. Based on the insights obtained from the analyses, we thus propose an RL-instructed LLM-based agent framework. This framework leverages a policy optimized by reinforcement learning (RL) to determine an appropriate discussion tactic (*e.g.*, "Honest Evidence", "Deceptive Accusation", etc.) based on current observation. To evaluate the effectiveness and generalizability of our framework, we conduct experiments in a three-player and a five-player ONUW game. The results indicate that the integration of our discussion policy can help LLM-based agents approximate PBEs more closely and improve the performance of LLM-based agents. Moreover, we observe the discussion policy trained by RL performs better than that by directly prompting LLM.

Our contributions can be summarized as follows: Firstly, we formulate the ONUW game as a Multi-Phase Extensive-Form Bayesian Game and provide theoretical analyses of a three-player ONUW game, demonstrating that players' utilities are only determined by their beliefs, which reveals the pivotal role that discussion plays in the ONUW game. Secondly, we develop an environment of the ONUW game, which is complicated due to the uncertainty caused by role changes. And we additionally contribute a dataset featuring players employing various discussion tactics in the ONUW game. Finally, inspired by the importance of discussion shown in our analyses, we propose an RL-instructed language agent framework, where an instructive discussion policy trained by RL is integrated, and a belief modeling method is employed to deduce the roles of players and generate actions based on its beliefs and discussion tactics.

## 2 Related Work

This section reviews literature in the following areas: the analyses of Mafia/Werewolf games and the design of AI agents for communication games. Additional related work is covered in Appendix B.

Mafia is a multi-player communication game, modeling a conflict between the mafias and civilians. Early research [26–28] has attempted to identify the optimal strategies for both citizens and mafias theoretically and thereby determine what setups lead to fair games. However, these works are based on strong assumptions such as random lynchings or meaningless debate, and restricted to simple scenarios like without detectives or removing most discussion. Werewolf is a reinvention of the Mafia game by introducing additional roles with special abilities. An $\varepsilon$-Nash equilibrium is found under the limitation of both villager-side and werewolf-side strategies [29]. In contrast, our analyses make few assumptions and strictly adhere to game rules, despite being on specific cases. Some researchers have attempted to develop Werewolf agents that rely on rule-based systems or talking templates [30, 31], while some rely on language models directly trained with Werewolf logs [32]. As for the *One Night Ultimate Werewolf* game, Eger and Martens [33] builds AI agents that can choose statements from a fixed set and designs a system that allows human players to play games with their AI agents.

Recently, researchers have tried to explore the potential of LLMs playing the Werewolf game [21–23]. Xu et al. [22] utilizes a policy to select candidate actions generated by LLMs, which can be seen as an adjustment for the action distribution of the LLM-based agent. Wu et al. [23] applies RL to train a *Thinker* module for complex logical analysis and strategic planning using structured language features,

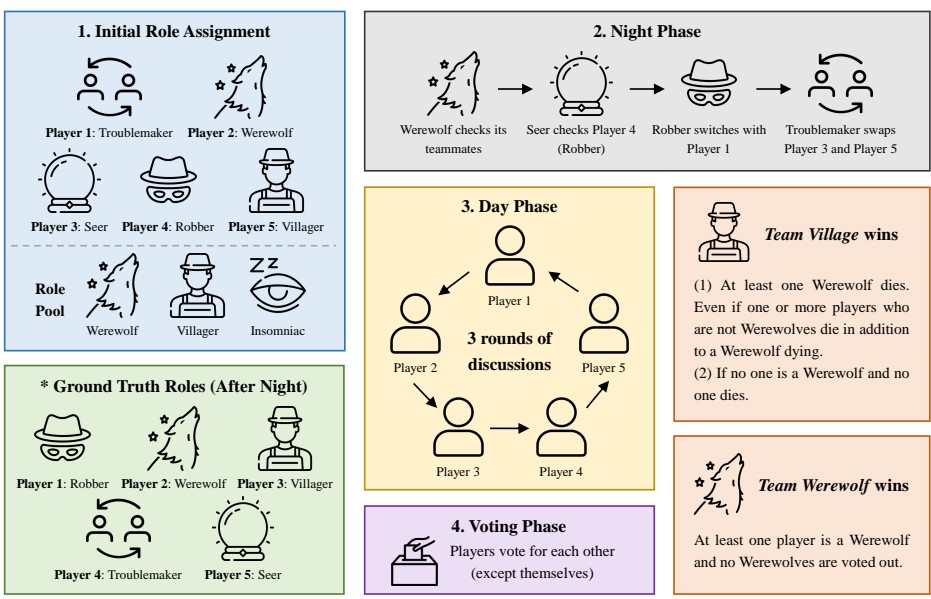

Figure 1: The game process of the ONUW game. Initially, roles are randomly dealt to players. Then three phases: Night (abilities performed in order), Day (discussion in three rounds), and Voting (suspicious player voted out) proceed sequentially. The winner is decided by the voting result.

aiming to improve the *System-2* reasoning ability of agents. In contrast, our work directly integrates a discussion policy trained by RL into the thinking and decision-making process of LLM-based agents, focusing on enhancing agents' strategic discussion (*e.g.*, honesty or deception) ability, which is an aspect that was not prominently addressed in previous work.

Besides the Werewolf games mentioned above, there are also AI researches conducted on other communication games, such as Diplomacy [3, 4], Avalon [18–20] and SpyFall [34, 35]. There has been a long history of studying Diplomacy in the realm of AI [36, 37], but most of them are rule-based algorithms until recently. Cicero [3] integrates a dialogue module with a planning module trained by RL to infer players' intentions and generate dialogue in pursuit of its plans, finally achieving human-level performance in Diplomacy. DeepRole [18] combines counterfactual regret minimization (CFR) with value networks trained through self-play, while introducing deductive reasoning techniques aimed at deducing actions within partially observable scenarios. Kim and Kim [34] demonstrates the LLMs' potential in playing a famous mafia-style game, SpyFall, by using prompt engineering techniques. Compared to our work, most designs of these agents are inadequate for handling the complex languages in the ONUW game and lack control over discussion tactics.

## 3 One Night Ultimate Werewolf Benchmark

*One Night Ultimate Werewolf* (ONUW) is a variant of the social game Werewolf [24]. In this game, players only have one night to use their abilities, followed by one day to discuss and win for their respective teams. The main challenges of this game are the role switches and potential deceptions that create uncertainty and confusion for all players. Initially, one role is dealt randomly to each player. There are three phases in the ONUW game: Night (abilities performed in order), Day (open discussion), and Voting (suspicious player voted out). Game rules are detailed in Appendix C.

### 3.1 Problem Formulation

Extensive-Form Game (EFG) [38, 39] is a type of game where players take turns to make decisions in a specific order. However, compared with classic EFG, there are three phases in the ONUW game, and each player possesses diverse action spaces that differ across these phases. Also, the private actions and hidden roles indicate players have incomplete information, which is usually modeled as a Bayesian Game (BG) [40, 41] or Hidden-Role Game (HRG) [42].

In order to formulate this game accurately, we consider a **Multi-Phase Extensive-Form Bayesian Game (MP-EFBG)**, whose representation is based on a game tree. An $m$-phase $n$-player MP-EFBG can be formalized as a tuple $(\mathcal{N}, \Psi, \mathcal{S}, \{\Theta^i\}_{i \in \mathcal{N}}, \{\mathcal{H}^i\}_{i \in \mathcal{N}}, \{I^i\}_{i \in \mathcal{N}}, \{\mathcal{A}^i_\psi\}_{i \in \mathcal{N}, \psi \in \Psi}, P, \Omega, R)$, where $\mathcal{N} = \{1, 2, \ldots, n\}$ is the set of players, and $\Psi = \{1, 2, \ldots, m\}$ is the set of phases. $\mathcal{S}$ denotes the set of states corresponding to nodes in the game tree. For each player $i$, $\Theta^i$ is the set of types and $\Theta = \times_{i=1}^n \Theta^i$ denotes the set of joint types of all players. To capture the private information, we define $\mathcal{H}^i$ as the set of information states and $I^i : \mathcal{S} \to \mathcal{H}^i$ as the information function that determines which states are indistinguishable for player $i$ by mapping them on the same information state. $\mathcal{A}^i_\psi(h^i)$ denotes the set of actions available to player $i$ at information state $h^i \in \mathcal{H}^i$ in phase $\psi \in \Psi$. So if two states $s_1, s_2 \in \mathcal{S}$ are mapped to the same information state by player $i$ ($I^i(s_1) = I^i(s_2)$), player $i$ will have the same action sets at these states. In this paper, we assume that all players have perfect recall, *i.e.*, each player's current information state $h^i_k$ implies knowledge of the sequence of its previous information states and actions, $(h^i_0, a^i_0, h^i_1, a^i_1, \ldots, h^i_k)$, that led to current information state. $P : \mathcal{S} \to \mathcal{N}$ is the player function, determining whom to act at the given state, and $\Omega : \mathcal{S} \to \Psi$ is the phase function that decides what phase it is at the given state. Finally, $R : \Theta \times \mathcal{S} \to \mathbb{R}^n$ is the utility function for all players according to their types and the terminal states (*i.e.*, leaf nodes in the game tree).

**Beliefs.** Since each player has incomplete information about other players' types and even its own type (player's role may be changed in the game), player $i$ will form a belief $b^i : \mathcal{H}^i \to \Delta(\Theta)$ on all players' types based on its observation. We define player $i$'s belief on information state $h^i \in \mathcal{H}^i$ as:

$$b^i(\theta|h^i) \stackrel{\text{def}}{=} \frac{p^i(\theta)p^i(h^i|\theta)}{\sum_{\theta' \in \Theta} p^i(\theta')p^i(h^i|\theta')} \tag{1}$$

where $p^i(\theta)$ is the prior probability of all players' types from player $i$'s view, and $p^i(h^i|\theta)$ is the probability that player $i$ observes $h^i$ given joint types $\theta$. Let $\Delta h^i_k = h^i_{k+1} \backslash h^i_k$ denote the new information contained in $h^i_{k+1}$ at step $k+1$ compared to $h^i_k$, then player $i$'s belief can be updated via Bayes' rule:

$$b^i_{k+1}(\theta|h^i_{k+1}) = \frac{b^i_k(\theta|h^i_k)p^i(\Delta h^i_k|\theta, h^i_k)}{\sum_{\theta' \in \Theta} b^i_k(\theta'|h^i_k)p^i(\Delta h^i_k|\theta', h^i_k)} \tag{2}$$

And $b^i_0(\theta|h^i_0) = b^i_0(\theta)$ is set as a uniform distribution, which is an unbiased estimate if there is no prior information.

**Behavioral Strategies.** Player $i$'s behavioral strategy in phase $\psi \in \Psi$ is defined as $\pi^i_\psi(h^i) \in \Delta(\mathcal{A}^i_\psi(h^i)), \forall h^i \in \mathcal{H}^i$, which is a probability distribution over available actions given a phase and an information state. And $\pi^i_\psi(a^i|h^i)$ denotes the probability that player $i$ takes action $a^i$ in phase $\psi$ given information state $h^i$. Player $i$'s strategy consists of belief modeling and action selection, and is expressed as

$$\pi^i_\psi(a^i|h^i) = \sum_{\theta \in \Theta} b^i(\theta|h^i)\tilde{\pi}^i_\psi(a^i|h^i, \theta) \tag{3}$$

where $\tilde{\pi}^i_\psi$ is a belief-conditioned behavioral strategy for player $i$ in phase $\psi$. Denote $\boldsymbol{\pi} = (\pi^1, \ldots, \pi^n)$ as a collection of strategies of all players, and the expected utility for player $i$ induced by strategy profile $\boldsymbol{\pi}$ as $\mathbb{E}_{\boldsymbol{\pi}}[R^i]$.

## 4 Analyses on Three-Player ONUW

In this section, we consider a simple but interesting version of the ONUW game: with two Werewolves and one Robber. We present certain Perfect Bayesian Equilibria [25] in this version of the game under two different assumptions, which demonstrate the crucial role that discussion during the Day phase plays in the ONUW game.

### 4.1 Notation

Since the ONUW game is an extensive-form game with incomplete information, an appropriate solution concept is Perfect Bayesian Equilibrium (PBE) $(\boldsymbol{\pi}, \boldsymbol{b})$: a strategy profile $\boldsymbol{\pi}$ and a belief system $\boldsymbol{b}$ satisfying *Belief Consistency* and *Sequential Rationality*. *Belief Consistency* includes both

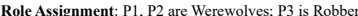

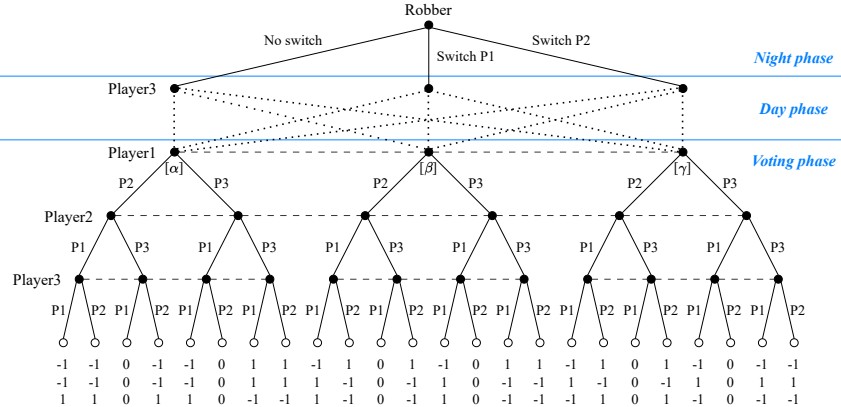

Figure 2: Game tree of the game with discussion. P1, P2, and P3 represent Player 1, Player 2, and Player 3, respectively. The dot lines in the Day phase represent Player 3's potential speeches. Those decision nodes on the same dash lines are in the same information sets for corresponding players. The utilities on leaf nodes are organized by the index of players.

on and off-equilibrium paths. Specifically, beliefs on every path reached in equilibrium with a positive probability (on-equilibrium path) should be updated according to Equation (2); on paths of zero probability (off-equilibrium path), the beliefs can be updated arbitrarily. For any player $i$ at given information state $h_k^i$, *Sequential Rationality* imposes the condition for strategy $\pi^i$: $\pi^i \in \arg\max_{\pi^{i\prime}} \mathbb{E}_{\pi^{i\prime}\pi^{-i}}\left[R^i|h_k^i, \boldsymbol{b}\right]$, which means given belief system and other players' subsequent strategies, player $i$'s strategy must be optimal.

Without loss of generality, we initially assign Player 1 and Player 2 as Werewolves, and Player 3 as Robber. There is only one Robber in the game which is known to all players. Denote Player 3's night behavioral strategy as Robber by $\pi_R^3$, and its voting strategies based on its night action as $\pi_{NS}^3$ (No switch), $\pi_{S1}^3$ (Switch with Player 1) and $\pi_{S2}^3$ (Switch with Player 2). Player 1 and Player 2 only have behavioral strategies in the Voting phase, denoting as $\pi^1, \pi^2$. Their beliefs in the Voting phase are $b^1, b^2$, and Player 3's beliefs at each information set are $b_{NS}^3, b_{S1}^3, b_{S2}^3$. Belief probability orders match decision node orders in the game trees (*e.g.*, Figure 2) from left to right. After the game, player $i$'s utility is $R^i = 1$ for winning, $R^i = -1$ for losing, $R^i = 0$ for drawing based on its final role.

### 4.2 Game without Discussion

First, we consider a scenario that discussion is not allowed in the Day phase, and players vote directly based on the private information they captured during the Night phase. Figure 6 shows the game tree in this case. At the beginning of the night, the Werewolves would recognize each other and thereby know the player left must be the Robber. However, based on the game rules, Robber always takes action after the Werewolves, which forms a threat to prevent them from directly voting for the Robber. Also, the Robber has no prior knowledge of two Werewolves, thus tends to switch with any of them with equal probability. The game's solution is PBEs expressed in the theorem below.

**Theorem 4.1.** *For the ONUW game with two Werewolves and one Robber, in the case where discussion is not allowed, there exist PBEs $(\boldsymbol{\pi}^*, \boldsymbol{b}^*)$: the Robber switches with any Werewolves with a probability of $1/2$ and votes for the player it switches with; the two Werewolves directly vote for each other. Each player's belief in $\boldsymbol{b}^*$ is consistent with other players' strategies. And the expected utilities of all players in the equilibria are:*

$$\mathbb{E}_{\boldsymbol{\pi}^*}\left[R^1\right] = \mathbb{E}_{\boldsymbol{\pi}^*}\left[R^2\right] = 0, \ \mathbb{E}_{\boldsymbol{\pi}^*}\left[R^3\right] = 1 \tag{4}$$

The formal description and proof of Theorem 4.1 can be seen in Appendix D.2. This theorem demonstrates that in PBEs of this case, the *original Robber* (*i.e.*, Player 3) always wins by randomly switching roles with the two other players to ensure its own safety and sow threats. Meanwhile, the probability of Player 1 and Player 2 winning is $1/2$ each, depending on whether they are switched by Player 3. This is because Player 3 knows all players' final roles due to its unique ability to switch.

### 4.3 Game with Discussion

Now we consider a scenario where discussion is allowed in the Day phase. If all players ignore the discussion, the PBEs in Theorem 4.1 will still hold. But a more common situation is that the beliefs held by two *original Werewolves* about *original Robber*'s night action would be influenced by Robber's speech, which would affect their voting strategies and result in different equilibria.

Based on Assumption D.1, we assume both *original Werewolves* would form beliefs about *original Robber*'s night action (No switch, Switch P1, Switch P2) with probabilities of $(\alpha, \beta, \gamma)$ after discussion, where $\alpha + \beta + \gamma = 1$. Figure 2 shows the game tree in this case. Then PBEs during the Voting phase can be derived based on the beliefs above.

**Theorem 4.2.** *For the ONUW game with two Werewolves and one Robber, in the case that both Werewolves form beliefs about the Robber's night action with probabilities of $(\alpha, \beta, \gamma)$ ($\alpha \neq 0$), there exist PBEs $(\pi^*, b^*)$ during the Voting phase: the Werewolves vote for Player 3 with a probability of $q$ and each other with $1 - q$, where $q = (\beta + \gamma - \alpha)/2\alpha$; the Robber votes for Player 1 with a probability of $p$ and Player 2 with $1 - p$, where $p = (\alpha^2 + \beta^2 - \gamma^2)/2\alpha^2$. Each player's belief in $b^*$ is consistent with other players' strategies. To ensure $p$ and $q$ are probabilities and the existence of the equilibria, there are constraints on the belief distribution (omitted for brevity). And under the constraints, the expected utilities of all players in these equilibria are:*

$$\mathbb{E}_{\pi^*}\left[R^1\right] = \delta(1 - 2\gamma), \ \mathbb{E}_{\pi^*}\left[R^2\right] = \delta(1 - 2\beta), \ \mathbb{E}_{\pi^*}\left[R^3\right] = -\delta \tag{5}$$

*where $\delta = 1/(4\alpha^2) - 1/(2\alpha) - 1$.*

The formal description and proof of Theorem 4.2 can be seen in Appendix D.3. Interestingly, Player 3's utility is only determined by the probability of the Werewolves believing it did not switch with anyone. Therefore, if Player 3 wants to maximize its utility through discussion while maintaining the equilibria, it has to convince the Werewolves that it did not switch with a probability of $1/2$, no matter what action it took. In this way, Player 3's expected utility achieves the maximum 1.

## 5 Learning to Discuss Strategically

In Section 4, we demonstrate that it is essential for players to affect other players' beliefs through discussion. As a result, we capture knowledge about how to discuss in a latent variable $Z$ (called discussion tactic), on which we condition the behavioral strategy with belief as $\tilde{\pi}_\psi^i(a|h, \theta, z)$, to explicitly adjust player's discussion preference in their strategy. Players decide their discussion tactic after being given their own information and derived beliefs. Hence, when introducing the discussion tactic $z \in Z$, the behavioral strategy of player $i$ (*i.e.*, Equation (3)) can be rewritten as:

$$\pi_\psi^i(a|h) = \sum_{\theta \in \Theta} b^i(\theta|h) \sum_{z \in Z} \mu^i(z|h, \theta) \tilde{\pi}_\psi^i(a|h, \theta, z) \tag{6}$$

where $\mu^i$ is denoted as the discussion policy of player $i$, deciding what specific discussion tactic $z$ to choose given information and belief. In the following, we describe our method for learning the discussion policy and how it is applied to construct an LLM-based agent to play the ONUW game.

### 5.1 Learning Discussion Policy by RL

Considering the remarkable reasoning and human-like text generation abilities of LLMs, we adopt LLMs as the belief function $b^i$ and the belief-conditioned behavioral strategy $\tilde{\pi}_\psi^i$ of player $i$. As a consequence, we only need to learn the discussion policy $\mu^i$ to obtain a better behavioral strategy for player $i$ according to Equation (6). Since the objective of learning policy $\mu^i$ is to maximize the expected payoffs of player $i$, we can optimize it in an RL manner if we treat other players' public actions (which are contained in player $i$'s information state $h^i$) and player $i$'s derived belief $\theta$ as its observation and the chosen discussion tactic $z^i$ as its action. Meanwhile, since players in the ONUW game only receive their rewards (or utilities) at the end of the game, the reward function at each step can be defined as follows: if game is over at the next step, then $r^i(h^i, \theta; z^i) = R^i(\theta^*, s)$, where $\theta^*$ is the ground truth of the role assignment; otherwise, $r^i(h^i, \theta; z^i) = 0$.

**Discretization of Discussion Tactic.** Due to the complexity of languages and semantics, there are almost unlimited types of discussion tactics. This makes it difficult to learn the policy $\mu^i$ and

understand the chosen discussion tactic $z$. Inspired by prior research on argumentation [43, 44] and the discussion characteristics of the ONUW game, we classify the discussion tactics frequently adopted during the game into three major categories:

- **Evidence**: *Provide some game-related evidence or information.* It is a widely used and main tactic in communication games for players.

- **Accusation**: *Accuse someone has a specific role or action.* Accusation is a generic tactic in games with hidden roles, which can force the accused player to reveal information to defend itself.

- **Defense**: *Defend yourself or someone else against an accusation.* Accompanied by accusation, there is defense. Player's defense can also be seen as a "reverse accusation" back to the accuser.

For each category, we further divide it into *honest* and *deceptive* ones (*e.g.*, "Honest Evidence" and "Deceptive Evidence"). Here, "honest" refers to being consistent with the information or beliefs that the speaker knows, while "deceptive" means the opposite. Hence, we get a total of six discussion tactics, which can be considered as an intuitive discretization of the discussion tactic space $Z$, enabling us to analyze the discussion preference of players explicitly.

**Optimizing Discussion Policy with RL.** Although LLMs can serve as the discussion policy $\mu^i$ for each player, the decision-making ability of LLMs on specific tasks tends to be affected by their prior knowledge of other tasks. In order to enhance their performance, we implement reinforcement learning to optimize a task-specific discussion policy for the ONUW game. Since there is no extraneous game information, such as players' personalities or body language, that could interfere with gameplay, it is believed that there exists an optimal policy $\mu$ that is invariant across different players. Therefore, we decide to train a general discussion policy $\mu$ that adapts to various situations.

In contrast to traditional RL tasks, the state for policy $\mu$ is the discussion history $h$ and derived belief on all players' roles $\theta$, which are both presented in natural language. To combine these into a single, fixed-length input, we concatenate the history and belief and convert them into state embeddings using LLMs: $s = \texttt{LLMEmbedding}(h, \theta)$, where we consider all necessary information is extracted. The state embedding $s$ is then passed through the discussion policy $\mu$, and the output is the chosen discussion tactic $z$. However, because of the slow interaction with LLMs, it is almost impossible to optimize the discussion policy $\mu$ using online RL methods. So we turn to offline RL methods that support optimizations on discrete action spaces.

Given the scarcity of datasets containing human players engaged in the ONUW game, we opt to leverage game logs generated by LLMs, which is the most effective way to collect trajectories for offline RL training. We first extract the trajectories separately from each player's perspective from the game logs. For each transition in these trajectories, the discussion history $h$ that is visible to the current player and the derived belief $\theta$ are converted to the current state embeddings $s$ by LLMs, so does the next state embeddings $s'$. Then these transitions are gathered to create a dataset $\mathcal{D} = \{(s, z, r, s')\}$ for the following training. We adopt Conservative Q-Learning (CQL) [45] to train the discussion policy $\mu$. Let $Q_\phi(s, z)$ denote the Q-function of policy $\mu$, parameterized by $\phi$, the loss function can be written as

$$\mathcal{L}(\phi) = \rho \mathbb{E}_{s \sim \mathcal{D}} \left[ \log \sum_z \exp Q_\phi(s, z) - \mathbb{E}_{z \sim \mathcal{D}} \left[ Q_\phi(s, z) \right] \right] + \frac{1}{2} \mathbb{E}_{s, z, s' \sim \mathcal{D}} \left[ \left( Q_\phi - \mathcal{B} Q_\phi \right)^2 \right] \quad (7)$$

where $\rho$ is a trade-off factor, and $\mathcal{B}$ is the Bellman operator.

## 5.2 RL-instructed LLM-based Agent Framework

Here we demonstrate how to utilize the trained discussion policy to construct an LLM-based agent to play the ONUW game. An overview of the framework is shown in Figure 3. There are three key components: (1) *Belief Modeling*; (2) *Discussion Tactic Selection*; and (3) *Decision Making*. Different combinations of the components are used to generate actions in different phases. For the Night phase (Component 3), since there is no additional information to refer to, players would take actions based solely on their observations and prior information. For the Day phase (Component 1, 2, 3), all components combine to generate a public speech that adheres to the discussion tactic chosen by the trained policy. And for the Voting phase (Component 1, 3), players would first form beliefs and then vote for one player based on their observations and beliefs (mostly).

In Component 1 and 3, where LLM is used to form beliefs and generate actions, we design the full prompts as follows: for *Belief Modeling* (Component 1), the system prompt consists of an

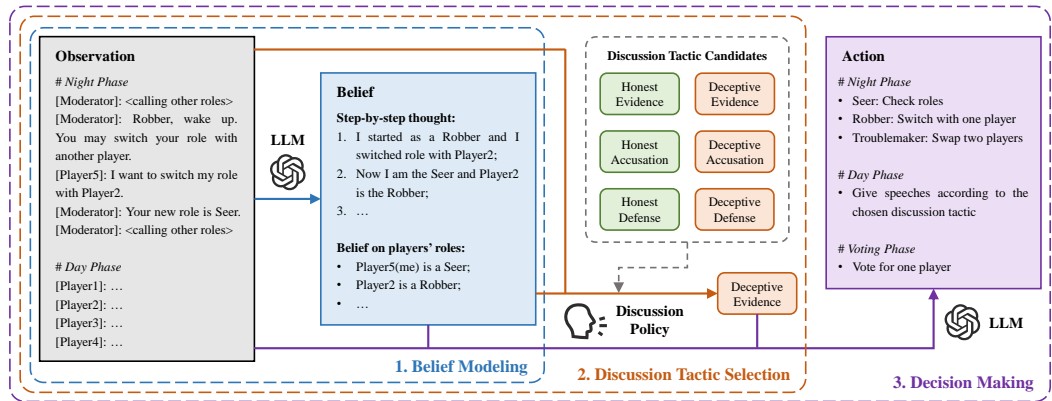

Figure 3: Overview of the RL-instructed LLM-based agent framework. (1) Belief Modeling: form beliefs on players' roles based on the observation. (2) Discussion Tactic Selection: utilize a discussion policy trained by RL to choose a discussion tactic from the candidates. (3) Decision Making: take action based on the observation (also belief and discussion tactic, according to the game phase).

explanation that LLM is playing the ONUW game, the rules of the game, all possible roles and their abilities, detailed description of its role, and the desired response format; the user prompt includes observation of the player that LLM is, and instructions about how to form beliefs. For *Decision Making* (Component 3), the system prompt adds additional prompts of the chosen discussion tactic; the user prompt includes observation, beliefs derived from the *Belief Modeling* component, and instructions about how to act in the current phase. Our prompts are detailed in Appendix H.

# 6 Experiments

In this section, we conduct three experiments from different aspects to evaluate the performance of our proposed RL-instructed LLM-based agent framework. We first perform experiments on the three-player ONUW game discussed in Section 4, to test whether LLM-based agents can recognize and approximate the equilibria. Then to showcase the scalability of our method, we consider a five-player ONUW game for experiments. We evaluate the effectiveness of our discussion policy trained by RL in settings where the initial role assignment and players' actions at night are predefined. Furthermore, we demonstrate the generalizability of our discussion policy in the standard setting to see whether it could adapt to any role in the game.

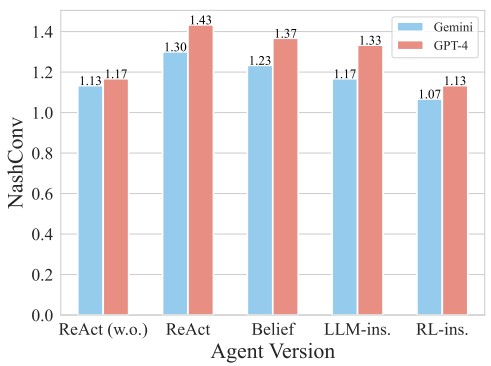

Figure 4: The NashConv value of different agents playing in the three-player ONUW game.

## 6.1 Setup

To develop the game environment, we employ a multi-agent language game framework called ChatArena [46] and modify it to fit the ONUW game. We conduct our experiments on `gpt-4-1106-preview` [3] (GPT-4) and `gemini-pro` [4] (Gemini), where the temperature is all set to 1.0. Considering the state-of-the-art performance of GPT-4 [47], we leverage the game logs generated by it as the dataset for training the discussion policy. Detailed collection process and data statistics are in Appendix E.1. `text-embedding-ada-002` [3] is adopted to get the state embeddings and we utilize the CQL [45] to train the discussion policy. More training details can be referred to Appendix E.2. We repeat the game 30 times and report the final results for each evaluation.

[3] https://platform.openai.com/docs/models
[4] https://ai.google.dev/models

As for comparison, we implement the *ReAct* [48] agent as the baseline by directly prompting the LLM with raw observations to generate its reasoning and action. Besides, we design three other ablated versions of our LLM-based agents (*RL-instructed*) to conduct evaluations. The first ablated version (*Belief*) removes the discussion policy, which means the agent generates speech directly based on its observations and beliefs. The second version (*Random*) adopts a random discussion policy. And the last version (*LLM-instructed*) replaces the discussion policy with LLM, allowing LLM to determine the discussion tactic autonomously.

## 6.2 Experiment on Three-Player ONUW

The analyses in Section 4 theoretically illustrate the PBEs in the three-player ONUW game. Therefore, in this section, we investigate the potential of LLM-based agents to recognize and approximate the equilibria by calculating the NashConv value of agents when playing in the same game setting. The NashConv value is defined as $\text{NASHCONV}(\pi) = \sum_i \left[ R^i(\text{BR}(\pi^{-i}), \pi^{-i}) - R^i(\pi) \right]$, where BR means the best response. This value represents how much utilities players can gain by deviating to their best responses, which also can be interpreted as a distance to equilibria.

We apply all four versions of agents to play in the three-player ONUW game while using Gemini and GPT-4 as the backend LLMs, respectively. And to demonstrate the impact of discussion on players approximating the equilibria, we also conduct experiments on *ReAct* agents playing the three-player ONUW game without discussion (denoted as *ReAct (w.o.)*). The results are shown in Figure 4. Firstly, the introduction of discussion in the

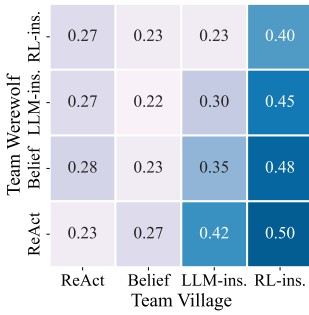 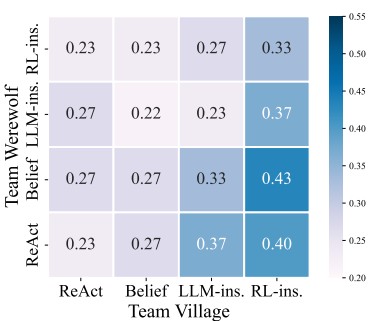

(a) *Team Village*'s win rates in the *easy* setting

(b) *Team Village*'s win rates in the *hard* setting

Figure 5: The matrices of *Team Village*'s win rates in different settings. It is clear that the hard setting weakens the advantages of our agent.

game increases the difficulty for players to approximate the equilibria. However, the discussion policy (*i.e.*, *LLM-instructed* and *RL-instructed*) can help LLM-based agents make better responses and thereby reduce their NashConv values. And in contrast, the discussion policy trained by RL performs better than directly prompting the LLM. It is interesting to notice that the NashConv values of GPT-4-based agents are higher than Gemini-based ones. Detailed analysis of evaluation logs reveals that it is because when employing Gemini, the two Werewolves are more likely to both vote for the Robber, making it unable to find a better response given their actions.

## 6.3 Effectiveness of the Discussion Policy

As our agent can better approximate the equilibria in the three-player ONUW game by integrating a discussion policy, we extend the experiments to a five-player version of the game to show its scalability. Since the discussion policy is the key component of our agent, we evaluate its effectiveness by letting our agent play as *Team Village* or *Team Werewolf* against other agents. We adopt two environments with different difficulty levels (*easy* and *hard*), where the initial role assignment and players' night actions are predefined. Compared to the *easy* setting, the Werewolf is switched by the Robber in the *hard* one, resulting in the rise of reasoning difficulty during the game. More detailed environment settings can be found in Appendix E.3.

Considering that the discussion policy is trained on the dataset generated by GPT-4, we employ Gemini as the backend LLM for all four versions of agents to better illustrate the result. The matrices of *Team Village*'s win rates under two different game settings are shown in Figure 5. Each column in the matrices corresponds to the win rates that all players on *Team Village* utilize the same version of agent against *Team Werewolf*, where all members also utilize the same version. As shown in each row of two matrices, our agent (playing as *Team Village*) always achieves the highest win rates when competing against the same versions of agents under both game settings. And it is similar for

each column. However, it is notable that the performance improvement of our agent is greater when playing as *Team Village* than *Team Werewolf* since the gap in win rates between agents is wider in *Team Village*. We argue it is possibly caused by the game mechanics where the Werewolves tend to have high win rates. Hence, our discussion policy showcases less significant improvement for *Team Werewolf* compared to *Team Village*.

### 6.4 Generalizability of the Discussion Policy

Since the discussion policy is trained to deal with various situations, we further carry out experiments in a standard five-player ONUW environment. However, due to the complexity of the role changes during the Night phase, players find it challenging to deduce their own final roles (except Insomniac) and some might even never figure it out. Therefore, we evaluate our RL-instructed LLM-based agent by setting it as one specific player while others as the other versions of agents. In this way, we demonstrate the generalizability of our discussion policy trained by RL, as it can assist the LLM-based agent in gaining better performance no matter what situation occurs to it.

Table 1: Win rates and average votes of our agents when playing the five-player ONUW game as Player 3.

| Agents | Gemini (*Belief*) | | GPT-4 (*Belief*) | |
|---|---|---|---|---|
| | Win Rates | Avg. Votes | Win Rates | Avg. Votes |
| *ReAct* | 0.40 | 1.23 | 0.30 | 1.73 |
| *Belief* | 0.40 | 1.73 | 0.32 | 1.87 |
| *Random* | 0.37 | 1.53 | 0.32 | 2.03 |
| *LLM-ins.* | 0.62 | 1.10 | 0.37 | 1.90 |
| *RL-ins.* | **0.70** | 1.10 | **0.50** | 1.87 |

Similarly, we conduct experiments with all five versions of agents and leverage Gemini as the backend LLM. Meanwhile, to reduce the possible impact from the player's index, we designate our agent to play the five-player ONUW game as Player 3. As for other players, we adopt two versions of agents: (1) Gemini-based *Belief* agent and (2) GPT-4-based *Belief* agent. We report the performance of our agents playing as Player 3 against other agents in Table 1. Considering the objective of discussion policy, which is to improve performance and meanwhile clarify or conceal themselves (*i.e.*, get fewer votes), we propose two metrics: *win rates* and *average votes*, to measure the generalizability of discussion policy when playing different roles. The results demonstrate that our agent outperforms other versions when playing against both Gemini-based and GPT-4-based agents. And it can even achieve a tie with GPT-4 in multiple rounds of evaluation.

## 7 Conclusion and Future Work

In this work, we propose a novel RL-instructed LLM-based agent framework that achieves outstanding performance in the *One Night Ultimate Werewolf* game. Addressing the ONUW game as a Multi-Phase Extensive-Form Bayesian Game, we highlight the pivotal role of discussion in players' strategies through certain Perfect Bayesian Equilibria in the game. Our experimental results in several settings demonstrate the effectiveness and generalizability of the discussion policy and our proposed agent framework. In general, this work provides new research insights for communication games with inherent uncertainty and the integration of reinforcement learning into LLM-based agents.

One major limitation of our approach is the manual discretization of discussion tactics, which heavily relies on the inherent characteristics within specific communication games. Therefore, as for future work, it is worth investigating how to extract discussion tactics of any communication games automatically based on game logs in an unsupervised fashion. Additionally, it would be interesting to explore the sensitivity of our agents when selecting different combinations of discussion tactics.

## Acknowledgements

We acknowledge the support from the National Natural Science Foundation of China (Grant No. 62206289), which supported Haifeng Zhang and Xuanfa Jin.

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

# Appendix

# A  Game Tree of the Game without Discussion

Here is the game tree of the game without discussion in Section 4.2.

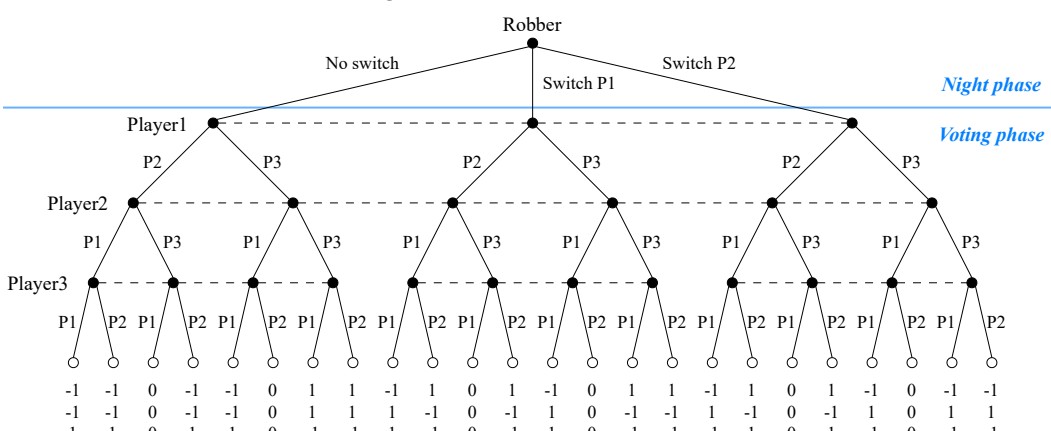

Figure 6: Game tree of the game without discussion. P1, P2, and P3 represent Player 1, Player 2, and Player 3, respectively. The decision nodes on the same dash lines are in the same information sets for corresponding players. The utilities on leaf nodes are organized by the index of players (*i.e.*, from top to bottom, they are the utilities of Player 1, Player 2, and Player 3).

# B  Additional Related Work

In this section, we review literature on the advancement of current LLM-based agents.

Due to the remarkable reasoning and emergent generalization abilities of LLMs, there has been a notable advancement in constructing LLM-based agents for various tasks, including sandbox and strategy games [15–17, 19, 49], social simulation [8, 50, 51], software development [52] and web operations [53, 7, 54], etc. Most of these works leverage LLMs as the "brain" of agents based on the ReAct [48] paradigm, which generates reasoning traces and actions in an interleaved manner to interact with the environment and thereby make decisions. Chain-of-Thought (CoT) [55] and Tree-of-Thought (ToT) [56] can improve the performance of LLM-based agents on solving complicated tasks by decomposing them into simpler ones. Specifically, CoT prompts LLMs to think step-by-step, while ToT creates a tree structure of different thoughts and searches for the best one.

# C  Detailed Introduction to the *One Night Ultimate Werewolf* Game

*One Night Ultimate Werewolf* (ONUW) is a variant of the social game *Werewolf*. In this game, players only have one night to use their abilities and one day to win for their teams. The challenge of this game is the role switch and potential deceptions that create uncertainty and confusion for all players.

## C.1  Game Setup

The setup of the ONUW game differs based on the number of players. There should always be three more roles than the number of players. At the beginning of the game, the roles are shuffled and one is dealt to each player while three extra roles are dealt to the role pool. After the role assignment, each player should secretly view their role and not reveal it to others. In our work, we mainly consider a five-player ONUW game with 8 roles: 2 Werewolves, 2 Villagers, 1 Seer, 1 Robber, 1 Troublemaker, and 1 Insomniac.

## C.2  Role Descriptions

Here we only describe the roles supported in our implementation of the game:

- **Werewolf**: At night, all Werewolves wake up and look for other Werewolves. If no one else wakes up, it means the other Werewolves are in the role pool. Werewolves are on *Team Werewolf*.

- **Villager**: The Villager does not wake up at night and has no special abilities or information. Players may often claim to be a Villager. The Villager is on *Team Village*.

- **Seer**: At night, the Seer may look either at one other player's role or at two roles in the role pool. The Seer is on *Team Village*.

- **Robber**: At night, the Robber may choose to switch a role with another player and then view its new role. The player who is switched is on *Team Village*. The Robber is on the team of its new role, however, it does not do the action of its new role at night. If the Robber chooses not to switch, it remains the Robber and is on *Team Village*.

- **Troublemaker**: At night, the Troublemaker may swap the roles of two other players without looking at their roles. The players who receive a different role are now on the team of their new role, even though they do not know what role that is until the end of the game. The Troublemaker is on *Team Village*.

- **Insomniac**: The Insomniac wakes up and looks at its role (to see if it has changed) at the end of the night. The Insomniac is on *Team Village*.

### C.3  Game Play and Winning Conditions

There are three phases in the ONUW game, including Night Phase, Day Phase, and Voting Phase.

- **Night Phase**: Several roles would be called on at night to do a night action, according to their initial roles. The calling order in our setting is (1) Werewolf, (2) Seer, (3) Robber, (4) Troublemaker, and (5) Insomniac.

- **Day Phase**: After the Night Phase, players discuss amongst themselves who they believe the Werewolves are. But Werewolves might want to claim a different role so that they do not die. Because certain roles change other players' roles, some players will believe they are one role, when they are actually a different one.

- **Voting Phase**: After several rounds of discussion during the Day phase, players vote for other players they believe are most likely to be Werewolf at the same time. The player with the most votes dies and reveals its role. In case of a tie, all players tied with the most votes die and reveal their roles. And if no player receives more than one vote, no one dies.

After just one Night, one Day, and one Voting Phase, the game ends. Here are the winning conditions:

- The *Team Village* wins: (1) If at least one Werewolf dies. Even if one or more players who are not Werewolves die in addition to a Werewolf dying, everyone on *Team Village* wins. (2) If no one is a Werewolf and no one dies. It is possible if all Werewolf roles are in the role pool.

- The *Team Werewolf* wins: If at least one player is a Werewolf and no Werewolves are voted out.

## D  Proofs for Perfect Bayesian Equilibria

Here we present the proofs for Perfect Bayesian Equilibria (PBEs) in Section 4.

### D.1  Notation

Without loss of generality, we assign roles to players at the beginning of the game as follows: Player 1 and Player 2 are the Werewolves, and Player 3 is the Robber. There is only one Robber in the game which is known to all players. Let $A_N^3$ denote the set of actions that Player 3 could take during the Night phase as a Robber, we have $A_N^3 = \{\text{No switch, Switch Player 1, Switch Player 2}\}$. As for the Voting phase, players' action sets vary according to their index: $A_V^1 = \{\text{Vote Player 2, Vote Player 3}\}$, $A_V^2 = \{\text{Vote Player 1, Vote Player 3}\}$, $A_V^3 = \{\text{Vote Player 1, Vote Player 2}\}$. For Player 3, we denote its behavioral strategy as a Robber in the Night phase as $\pi_R^3 \in \Delta(A_N^3)$. And Player 3 has three different information sets during the Voting phase depending on its action at night, thus there are three behavioral strategies: $\pi_{NS}^3, \pi_{S1}^3, \pi_{S2}^3 \in \Delta(A_V^3)$, corresponding to cases where Player 3 did not switch,

switched with Player 1 and with Player 2. And for Player 1 and Player 2, they only have behavioral strategies during the Voting phase, which are defined as $\pi^1 \in \Delta(A_V^1), \pi^2 \in \Delta(A_V^2)$.

Regarding beliefs, we denote the beliefs that Player 1 and Player 2 hold when they are in the Voting phase as $b^1, b^2$. Meanwhile, let $b_{NS}^3, b_{S1}^3, b_{S2}^3$ denote the beliefs that Player 3 holds when it is at different information sets in the Voting phase. In subsequent analyses, the order of probabilities in all beliefs is consistent with that of decision nodes in their corresponding information sets in the game trees (*e.g.*, Figure 2) from left to right. After the game ends, the winning condition depends on the final role of each player. If player $i$ wins, its utility $R^i = 1$; if loses, $R^i = -1$; and if draws, $R^i = 0$.

From the game tree in Figure 2 and Figure 6, we find the weakly dominant strategies for $\pi_{S1}^3$ and $\pi_{S2}^3$:

$$\pi_{S1}^{3,*} = (1,0), \ \pi_{S2}^{3,*} = (0,1) \tag{8}$$

Since it is hard to directly calculate the PBEs of any given game, we leverage the necessary condition of PBEs that *Any PBEs of a game must be the Nash Equilibria (NEs)*, so we first calculate the NEs of a game and then verify whether there exist PBEs in them. And to simplify the calculation, we adopt Equation (8) as part of NEs in subsequent proofs.

**Assumption D.1.** Assume that the two *original Werewolves* are homogeneous, which means there is no difference between them and so are their belief models.

## D.2 Proof for Theorem 4.1

**Theorem D.2.** *For the ONUW game with two Werewolves and one Robber, in the case where discussion is not allowed, there exist PBEs $(\boldsymbol{\pi}^*, \boldsymbol{b}^*)$:*

$$\begin{cases} \pi^{1,*} = (1,0) \\ \pi^{2,*} = (1,0) \\ \pi_R^{3,*} = \left(0, \frac{1}{2}, \frac{1}{2}\right) \\ \pi_{NS}^{3,*} = (p, 1-p) \\ \pi_{S1}^{3,*} = (1,0) \\ \pi_{S2}^{3,*} = (0,1) \end{cases} \tag{9} \qquad \begin{cases} b^{1,*} = \left(0, \frac{1}{2}, \frac{1}{2}\right) \\ b^{2,*} = \left(0, 0, \frac{1}{2}, 0, \frac{1}{2}, 0\right) \\ b_{NS}^{3,*} = (1,0,0,0) \\ b_{S1}^{3,*} = (1,0,0,0) \\ b_{S2}^{3,*} = (1,0,0,0) \end{cases} \tag{10}$$

*where $p \in [0,1]$, hence there are infinite equilibria in this case. And the expected utilities of all players in these equilibria are:*

$$\mathbb{E}_{\boldsymbol{\pi}^*}\left[R^1\right] = \mathbb{E}_{\boldsymbol{\pi}^*}\left[R^2\right] = 0, \ \mathbb{E}_{\boldsymbol{\pi}^*}\left[R^3\right] = 1 \tag{11}$$

*Proof.* Considering the symmetry of Player 1 and Player 2 in the game without discussion, we assume that they share the same probability of voting for Player 3, and the Robber (*i.e.*, Player 3) switches with any of them with the same probability. Therefore, we can set

$$\pi_R^3 = (1-2s, s, s), \ \pi^1 = (1-q, q), \ \pi^2 = (1-q, q), \ \pi_{NS}^3 = (p, 1-p) \tag{12}$$

where $0 \le s \le 1/2$ and $0 \le p, q \le 1$. Based on the given strategy profile $\boldsymbol{\pi}$ and the utilities on the game tree Figure 6, the expected utilities of each player can be written as:

$$\mathbb{E}_{\boldsymbol{\pi}}\left[R^1\right] = (1-2s)\left(q^2 + q - 1\right) \tag{13}$$

$$\mathbb{E}_{\boldsymbol{\pi}}\left[R^2\right] = (1-2s)\left(q^2 + q - 1\right) \tag{14}$$

$$\mathbb{E}_{\boldsymbol{\pi}}\left[R^3\right] = -q^2 - q + 1 \tag{15}$$

According to the definition of NEs, if the strategy profile $\boldsymbol{\pi}$ is a NE, it can be derived that:

$$\frac{\partial \mathbb{E}_{\boldsymbol{\pi}}\left[R^1\right]}{\partial q} = \frac{\partial \mathbb{E}_{\boldsymbol{\pi}}\left[R^2\right]}{\partial q} = (1-2s)(2q+1) = 0 \tag{16}$$

$$\frac{\partial \mathbb{E}_{\boldsymbol{\pi}}\left[R^3\right]}{\partial s} = 0 \tag{17}$$

$$\frac{\partial \mathbb{E}_{\boldsymbol{\pi}}\left[R^3\right]}{\partial p} = 0 \tag{18}$$

So when $s^* = 1/2$, the strategy profiles $\boldsymbol{\pi}$ in Equation (12) become NEs of this game.

To find out possible PBEs in the NEs above, we form beliefs $\boldsymbol{b}$ that satisfy the *Bayesian Consistency* requirement:

$$b^1 = \left(0, \frac{1}{2}, \frac{1}{2}\right), \; b^2 = \left(0, 0, \frac{1-q}{2}, \frac{q}{2}, \frac{1-q}{2}, \frac{q}{2}\right),$$
$$b_{\text{NS}}^3 = b_{\text{S1}}^3 = b_{\text{S2}}^3 = \left((1-q)^2, (1-q)q, (1-q)q, q^2\right) \tag{19}$$

Then we need to figure out the exact $p$ and $q$ that make strategies meet the *Sequential Rationality* requirement. Let $\mathbb{E}^i[a], a \in \mathcal{A}^i(h)$ denote the expected utility of player $i$ taking action $a$ at information set $h$. Therefore, we analyze the best response of each player at each information set as follows:

- **For Player 3 given belief $b_{\text{NS}}^3$:**

$$\mathbb{E}^3[\text{P1}] = (1-q)^2 + (1-q)q - q^2 = -q^2 - q + 1,$$
$$\mathbb{E}^3[\text{P2}] = (1-q)^2 + (1-q)q - q^2 = -q^2 - q + 1 \tag{20}$$

  It indicates Player 3 can take any mixed behavioral strategy, which means $p$ could be any value between 0 and 1.

- **For Player 2 given belief $b^2$:** Note that when calculating the expected utilities of player's each action, the other players' subsequent strategies should be optimal given current beliefs. So following the best responses of Player 3, we have

$$\mathbb{E}^2[\text{P1}] = \frac{1-q}{2} + \frac{q}{2} - \frac{1-q}{2} = \frac{q}{2},$$
$$\mathbb{E}^2[\text{P3}] = -\frac{q}{2} - \frac{1-q}{2} + \frac{q}{2} = \frac{q-1}{2} \tag{21}$$

  Obviously, $\mathbb{E}^2[\text{P1}] > \mathbb{E}^2[\text{P3}]$ no matter what value $q \in [0, 1]$ is. So the best response of Player 2 is voting for Player 1, which is equal to $q = 0$.

- **For Player 1 given belief $b^1$:** Similarly, considering the best responses of Player 2 and Player 3, we have

$$\mathbb{E}^1[\text{P2}] = -\frac{1}{2} + \frac{1}{2} = 0, \; \mathbb{E}^1[\text{P3}] = -\frac{1}{2} \tag{22}$$

  Hence the best response of Player 1 is voting for Player 2, which is equal to $q = 0$, consistent with the result derived from Player 2's best response.

- **For Player 3 during the Night phase:**

$$\mathbb{E}^3[\text{No switch}] = 1, \; \mathbb{E}^3[\text{Switch P1}] = 1, \; \mathbb{E}^3[\text{Switch P2}] = 1 \tag{23}$$

  So strategy $\pi_{\text{R}}^3 = \left(0, \frac{1}{2}, \frac{1}{2}\right)$ is one of the best responses of Player 3 as a Robber during the Night phase.

The analysis above verifies that the following strategies and beliefs profile $(\boldsymbol{\pi}^*, \boldsymbol{b}^*)$ form the PBEs of the game:

$$\boldsymbol{\pi}^* = \left[\pi_{\text{R}}^{3,*}, \pi^{1,*}, \pi^{2,*}, \pi_{\text{NS}}^{3,*}, \pi_{\text{S1}}^{3,*}, \pi_{\text{S2}}^{3,*}\right] = \left[\left(0, \frac{1}{2}, \frac{1}{2}\right), (1,0), (1,0), (p, 1-p), (1,0), (0,1)\right] \tag{24}$$

$$\boldsymbol{b}^* = \left[b^{1,*}, b^{2,*}, b_{\text{NS}}^{3,*}, b_{\text{S1}}^{3,*}, b_{\text{S2}}^{3,*}\right] = \left[\left(0, \frac{1}{2}, \frac{1}{2}\right), \left(0, 0, \frac{1}{2}, 0, \frac{1}{2}, 0\right), (1,0,0,0), (1,0,0,0), (1,0,0,0)\right] \tag{25}$$

where $p \in [0, 1]$. So Theorem 4.1 is proved. $\qquad\square$

### D.3 Proof for Theorem 4.2

**Theorem D.3.** *For the ONUW game with two Werewolves and one Robber, in the case that both Werewolves form beliefs about the Robber's night action with probabilities of $(\alpha, \beta, \gamma)$ ($\alpha \neq 0$), there exist PBEs $(\pi^*, b^*)$ during the Voting phase:*

$$\begin{cases} \pi^{1,*} = (1-q, q) \\ \pi^{2,*} = (1-q, q) \\ \pi^{3,*}_{NS} = (p, 1-p) \\ \pi^{3,*}_{S1} = (1, 0) \\ \pi^{3,*}_{S2} = (0, 1) \end{cases} \quad (26) \quad \begin{cases} b^{1,*} = (\alpha, \beta, \gamma) \\ b^{2,*} = (\alpha(1-q), \alpha q, \beta(1-q), \beta q, \gamma(1-q), \gamma q) \\ b^{3,*}_{NS} = ((1-q)^2, (1-q)q, (1-q)q, q^2) \\ b^{3,*}_{S1} = ((1-q)^2, (1-q)q, (1-q)q, q^2) \\ b^{3,*}_{S2} = ((1-q)^2, (1-q)q, (1-q)q, q^2) \end{cases} \quad (27)$$

*where $q = (\beta + \gamma - \alpha)/2\alpha$, $p = (\alpha^2 + \beta^2 - \gamma^2)/2\alpha^2$. To ensure $p$ and $q$ are probabilities and the existence of the equilibria, there are constraints on the belief distribution:*

$$\begin{cases} \alpha + \beta + \gamma = 1 \\ \dfrac{1}{4} \leq \alpha \leq \dfrac{1}{2} \\ \dfrac{1 - 2\alpha}{2 - 2\alpha} \leq \gamma \leq \dfrac{2\alpha^2 - 2\alpha + 1}{2 - 2\alpha} \end{cases} \quad (28)$$

*Under the constraints above, the expected utilities of all players in these equilibria are:*

$$\mathbb{E}_{\pi^*}\left[R^1\right] = \left(\frac{1}{4\alpha^2} - \frac{1}{2\alpha} - 1\right)(1 - 2\gamma) \quad (29)$$

$$\mathbb{E}_{\pi^*}\left[R^2\right] = \left(\frac{1}{4\alpha^2} - \frac{1}{2\alpha} - 1\right)(1 - 2\beta) \quad (30)$$

$$\mathbb{E}_{\pi^*}\left[R^3\right] = -\left(\frac{1}{4\alpha^2} - \frac{1}{2\alpha} - 1\right) \quad (31)$$

*Proof.* Assuming both Player 1 and Player 2 form beliefs about Player 3's night actions with probabilities of $(\alpha, \beta, \gamma)$, based on Player 3's speech in the game with discussion. Since now the beliefs on the probability of Player 3 switching with Player 1 may not be equal to that with Player 2 (*i.e.*, there might be $\beta \neq \gamma$), the probabilities of Player 1 and Player 2 voting for Player 3 is no longer the same. In order to find the PBEs during the Voting phase in this case, we set

$$\pi^1 = (1 - q_1, q_1), \ \pi^2 = (1 - q_2, q_2), \ \pi^3_{NS} = (p, 1 - p) \quad (32)$$

where $0 \leq q_1, q_2, p \leq 1$. Similarly, we can get the expected utilities of each player based on the beliefs that Player 1 and Player 2 hold:

$$\mathbb{E}_{\pi}\left[R^1\right] = \alpha[q_1 q_2 + (q_2 - q_1)p + q_1 - 1] + \beta(q_1 q_2 + q_2 - 1) - \gamma(q_1 q_2 + q_1 - 1) \quad (33)$$

$$\mathbb{E}_{\pi}\left[R^2\right] = \alpha[q_1 q_2 + (q_2 - q_1)p + q_1 - 1] - \beta(q_1 q_2 + q_2 - 1) + \gamma(q_1 q_2 + q_1 - 1) \quad (34)$$

$$\mathbb{E}_{\pi}\left[R^3\right] = -\alpha[q_1 q_2 + (q_2 - q_1)p + q_1 - 1] - \beta(q_1 q_2 + q_2 - 1) - \gamma(q_1 q_2 + q_1 - 1) \quad (35)$$

For each player, in order to achieve the NEs, their strategies must satisfy the following conditions:

$$\frac{\partial \mathbb{E}_{\pi}\left[R^1\right]}{\partial q_1} = (\alpha + \beta - \gamma)q_2 - \alpha p + \alpha - \gamma = 0 \quad (36)$$

$$\frac{\partial \mathbb{E}_{\pi}\left[R^2\right]}{\partial q_2} = (\alpha - \beta + \gamma)q_1 + \alpha p - \beta = 0 \quad (37)$$

$$\frac{\partial \mathbb{E}_{\pi}\left[R^3\right]}{\partial p} = \alpha(q_1 - q_2) = 0 \quad (38)$$

Based on Equation (38), there are three possible cases: (a) $\alpha = 0, q_1 = q_2$; (b) $\alpha = 0, q_1 \neq q_2$ and (c) $\alpha \neq 0, q_1 = q_2$.

For case (a), it is under the same situation of Theorem 4.1, so PBEs in that theorem also hold in the current case. The results are omitted for brevity.

For case (b), we can derive from Equation (36) and Equation (37) that:

$$(\gamma - \beta)q_1 = \beta, \ (\beta - \gamma)q_2 = \gamma \tag{39}$$

However, it is impossible to keep both $q_1$ and $q_2$ non-negative while $\beta \neq \gamma$, as $\beta = \gamma$ is invalid.

For case (c), we can derive that:

$$q_1 = q_2 = q^* = \frac{\beta + \gamma - \alpha}{2\alpha}, \ p^* = \frac{\alpha^2 + \beta^2 - \gamma^2}{2\alpha^2} \tag{40}$$

Also, to keep both $p^*$ and $q^*$ being probabilities, there are constraints for belief $(\alpha, \beta, \gamma)$:

$$\begin{cases} \alpha > 0 \\ 0 \leq \beta + \gamma - \alpha \leq 2a \\ 0 \leq \alpha^2 + \beta^2 - \gamma^2 \leq 2a^2 \\ \alpha + \beta + \gamma = 1 \end{cases} \implies \begin{cases} \alpha + \beta + \gamma = 1 \\ \dfrac{1}{4} \leq \alpha \leq \dfrac{1}{2} \\ \dfrac{1 - 2\alpha}{2 - 2\alpha} \leq \gamma \leq \dfrac{2\alpha^2 - 2\alpha + 1}{2 - 2\alpha} \end{cases} \tag{41}$$

Similarly, we can form beliefs $b$ that satisfy the *Bayesian Consistency* requirement for all players, based on the result derived from NEs:

$$b^1 = (\alpha, \beta, \gamma), \ b^2 = (\alpha(1 - q^*), \alpha q^*, \beta(1 - q^*), \beta q^*, \gamma(1 - q^*), \gamma q^*),$$
$$b_{\text{NS}}^3 = b_{\text{S1}}^3 = b_{\text{S2}}^3 = \left((1 - q^*)^2, (1 - q^*)q^*, (1 - q^*)q^*, q^{*2}\right) \tag{42}$$

To further check whether the strategies derived from NEs above satisfy the **Sequential Rationality** requirement, we also analyze the best responses of each player at each information set during the Voting phase as follows:

- **For Player 3 given belief $b_{\text{NS}}^3$**: There are

$$\mathbb{E}^3[\text{P1}] = (1 - q^*)^2 + (1 - q^*)q^* - q^{*2} = -q^{*2} - q^* + 1,$$
$$\mathbb{E}^3[\text{P2}] = (1 - q^*)^2 + (1 - q^*)q^* - q^{*2} = -q^{*2} - q^* + 1 \tag{43}$$

  It shows any mixed behavioral strategies of Player 3 at this information set is optimal, so strategy $\pi_{\text{NS}}^3 = (p^*, 1 - p^*)$ is the best response.

- **For Player 2 given belief $b^2$**: Following the best responses of Player 3, we can derive:

$$\mathbb{E}^2[\text{P1}] = -\alpha(1 - q^*) - \alpha q^* p^* + \beta(1 - q^*) + \beta q^* - \gamma(1 - q^*)$$
$$= \left(\frac{1}{4\alpha^2} - \frac{1}{2\alpha} - 1\right)(1 - 2\beta),$$
$$\mathbb{E}^2[\text{P3}] = -\alpha(1 - q^*)(1 - p^*) + \alpha q^* - \beta q^* - \gamma(1 - q^*) + \gamma q^*$$
$$= \left(\frac{1}{4\alpha^2} - \frac{1}{2\alpha} - 1\right)(1 - 2\beta) \tag{44}$$

  Hence, any mixed behavioral strategies are also best responses, so does the strategy $\pi^2 = (1 - q^*, q^*)$.

- **For Player 1 given belief $b^1$**: Considering the best responses of Player 2 and Player 3, we have:

$$\mathbb{E}^1[\text{P2}] = \alpha(p^* q^* - 1) + \beta(q^* - 1) + \gamma = \left(\frac{1}{4\alpha^2} - \frac{1}{2\alpha} - 1\right)(1 - 2\gamma),$$
$$\mathbb{E}^1[\text{P3}] = \alpha(p^* q^* - p^* + q^*) + \beta(2q^* - 1) - \gamma q^* = \left(\frac{1}{4\alpha^2} - \frac{1}{2\alpha} - 1\right)(1 - 2\gamma) \tag{45}$$

  which also showcase the strategy $\pi^1 = (1 - q^*, q^*)$ is the best response of Player 1 given belief $b^1 = (\alpha, \beta, \gamma)$ and other players best responses.

Therefore, the analysis above demonstrates the strategies $\boldsymbol{\pi}^*$ and beliefs in Equation (42) form the PBEs during the Voting phase of the game:

$$\boldsymbol{\pi}^* = \left[\pi^{1,*}, \pi^{2,*}, \pi^{3,*}_{\text{NS}}, \pi^{3,*}_{\text{S1}}, \pi^{3,*}_{\text{S2}}\right] = [(1-q^*, q^*), (1-q^*, q^*), (p^*, 1-p^*), (1,0), (0,1)] \quad (46)$$

where $p^*, q^*$ follow Equation (40), while $(\alpha, \beta, \gamma)$ is under the constraints of Equation (41). So Theorem 4.2 is proved. □

## E Implementation and Experiment Details

### E.1 Data Collection and Statistics

In our experiment, we leverage GPT-4 to play the five-player ONUW games on 40 different game settings (*i.e.*, the role assignments) and repeat the games 3 times on each setting to gather diverse data, due to the randomness of the initial role assignment and the text generation of LLMs. Hence, we collect 120 game logs, each containing 15 discussion turns, resulting in a total of 1800 transitions that can be used for training.

To explore the GPT-4's preference for being honest or deceptive when playing different roles in the ONUW game, we analyzed the percentage of different discussion tactics that GPT-4 selects for each initial role in our dataset, as shown in Table 2. The results indicate that GPT-4 tends to be deceptive most of the time when it plays on *Team Werewolf* and honest when on *Team Village*, which aligns with the game objective and intuition. Notably, the Robber and Insomniac have a higher frequency of being deceptive than other players on *Team Village*. This is because they are the only two roles that can check their roles again, and if they see themselves becoming Werewolves, then may choose to be deceptive during the discussion. It is also interesting to note that GPT-4 rarely defends itself. We speculate that this is because accusing others can help transfer contradictions more effectively.

Table 2: Statistics on game logs generated by GPT-4. Here the **E**, **A** and **D** represent the discussion tactic of **Evidence**, **Accusation** and **Defense**, respectively.

| Initial Role | Honest | | | Deceptive | | |
|---|---|---|---|---|---|---|
| | E(%) | A(%) | D(%) | E(%) | A(%) | D(%) |
| Werewolf | 9.3 | 11.9 | 3.6 | 39.1 | 24.2 | 11.9 |
| Villager | 60.3 | 33.1 | 4.3 | 0.5 | 0.9 | 0.9 |
| Seer | 74.3 | 23.0 | 1.1 | 1.1 | 0 | 0.5 |
| Robber | 56.8 | 18.4 | 2.1 | 12.0 | 6.4 | 4.3 |
| Troublemaker | 75.8 | 21.3 | 1.4 | 0.5 | 0.5 | 0.5 |
| Insomniac | 70.4 | 16.9 | 1.9 | 4.1 | 3.7 | 3.0 |

### E.2 Offline RL Training and Hyperparameters

After collecting the game logs, we need to transfer them into the form that can be used for RL training. For each player in each game log, we extract the game history that is visible to it and the derived belief as its observation, and the selected discussion tactic as its action at each decision step. For the convenience of training, we use `text-embedding-ada-002` to convert the player's observations into state embeddings and turn its actions into indexes through a specific conversion when creating the dataset for offline RL training. The reason we adopt a frozen encoder for state embedding is that GPT's embedding model is well pretrained and we believe it could effectively represent the information of the original text in the semantic space. As for the reward, if the player wins, every decision step in this game gains 1; if loses, gains -1; and if draws, gains 0.

After creating the dataset, we use CQL [45], implemented by the d3rlpy library [57], to train the discussion policy for our RL-instructed LLM-based agent. The hyperparameters we used for CQL are listed in Table 3 (if not listed, use the default values).

Table 3: Training hyperparameters for CQL.

| Hyper-parameters | Value |
|---|---|
| Learning rate | 5e-5 |
| Discount factor ($\gamma$) | 0.99 |
| Mini-batch size | 32 |
| Trade-off factor ($\rho$) | 4.0 |
| Critic num | 2 |
| Target critic update interval | 1000 |
| Epoch num | 100 |
| Step num per epoch | 5000 |
| State dim | 1536 |
| Action dim | 6 |

Table 4: Role changes in *easy* setting.

| | Initial Role | Final Role |
|---|---|---|
| Player 1 | Troublemaker | Robber |
| Player 2 | Werewolf | Werewolf |
| Player 3 | Seer | Villager |
| Player 4 | Robber | Troublemaker |
| Player 5 | Villager | Seer |

Table 5: Role changes in *hard* setting.

| | Initial Role | Final Role |
|---|---|---|
| Player 1 | Robber | Werewolf |
| Player 2 | Insomniac | Seer |
| Player 3 | Seer | Insomniac |
| Player 4 | Werewolf | Robber |
| Player 5 | Troublemaker | Troublemaker |

### E.3 Game Settings in Section 6.3

To evaluate the effectiveness of the discussion policy trained by RL, we select two representative game settings from the five-player ONUW game, namely *easy* and *hard*, according to their difficulty level evaluated by humans. Players' night actions in both settings are predefined to make sure that players on the same team (finally) adopt the same version of agents. Table 4 and Table 5 show the initial and final roles of all players in both settings.

Players' actions during the Night phase in both settings are as follows: (1) **Easy setting**: First, the Seer (Player 3) checks Player 4, knowing it is the Robber. Then, the Robber (Player 4) switches roles with Player 1, becoming the new Troublemaker. At last, the Troublemaker (Player 1) swaps the roles of Player 3 and Player 5. (2) **Hard setting**: First, the Seer (Player 3) checks Player 4, knowing it is the Werewolf. Then, the Robber (Player 1) switched roles with Player 4, becoming the new Werewolf. At last, the Troublemaker (Player 5) swaps the roles of Player 2 and Player 3. Compared to the *easy* setting, the Werewolf is switched by the Robber in the *hard* one. Therefore, the new Werewolf would tend to conceal its actual role while the initial Werewolf might be wrongly deduced by other players, which increases the reasoning difficulty during the game.

### E.4 Computational Consumption

The computational consumption of implementing our RL-instructed LLM-based agent mainly consists of the compute workers for training the discussion policy by offline RL and the token usage of LLMs API. The training of the discussion policy takes an NVIDIA GeForce RTX 3060 Ti GPU for about 2.5 hours. As for the token usage, taking a five-player version of the ONUW game for example, playing one game costs approximately 70,000 to 80,000 tokens, which equals about 0.8 to 1.2 dollars if using GPT-4 as the backend LLM.

## F  Human Analysis

In this section, we present several qualitative results between our agent (*RL-ins.* agent) and *Belief* agent and provide corresponding human analyses on these results.

## F.1 Bluffing

> **Current situation**
>
> Our agent is Player 1, who is originally the Robber. During the Night phase, it switched its role with Player 4 and became the new Werewolf.

> *RL-ins.* agent
>
> - **Thought**: My role is Werewolf now, but I would like all players to think I am the Seer. Since I know no one is Werewolf except me, I should be safe if I don't reveal myself. But I want to know if the Insomniac exists and who the Insomniac is. So I will provide misleading evidence to induce the Insomniac to say his true action.
> - **Discussion tactic**: deceptive evidence
> - **Speech**: Hey everyone, I checked player2's role last night. He is not the Werewolf.

> *Belief* agent
>
> - **Thought**: 1. Player4 is very likely the new Robber. Because after I switched my role with player4, I became a Werewolf. So player4 must become the new Robber. 2. I am very likely the new Werewolf now, based on (1). 3. ... (omitted for brevity)
> - **Speech**: I switched my role with player4 in the Night phase. So player4 is very likely the Robber now.

> **Human analysis**
>
> - **Bluffing** is an advanced deceptive strategy often used in the ONUW game, involving deliberately deceiving or misleading others about players' real roles or actions. It carries a high risk of being caught, which can lead to loss of credibility. So it should be used carefully when giving deceptive information.
> - In this case, Player 1 knows it is the new Werewolf so it tends to bluff empirically to conceal its actual role. Our agent (*RL-ins.* agent) did bluff and chose to declare itself as the Seer, which is the most commonly declared role by our agent. Moreover, it declared that player 2 was not a Werewolf, instead of someone else being a Werewolf, which earned it support to resist questioning from others. However, when it comes to *Belief* agent, it was more likely to state its real night action while trying to hide its new role.

## F.2 Concealment

> **Current situation**
>
> Our agent is Player 3, who is originally the Seer. During the Night phase, it checked Player 4, knowing it was the Robber (originally).

> *RL-ins.* agent
>
> - **Thought**: I plan to mislead the players by falsely claiming that player 2 checked player 5 last night. This will potentially cause suspicion towards player 2, and may lead to a Werewolf being voted out. Additionally, I will avoid revealing my true role as the Seer, as this could put a target on my back.
> - **Discussion tactic**: deceptive accusation
> - **Speech**: I have a suspicion that player 2 is not being truthful about his role. I believe he may have checked player 5 last night. Let's keep an eye on player 2 and see if his actions match his words.

> **_Belief_ agent**
>
> - **Thought**: 1. Since player1 claimed that he swapped player3 and player5's roles, if it's true, it means player3 is not the Seer anymore and player5 is not the Troublemaker anymore. 2. However, player1 is acting suspiciously because he is the one who claimed to swap player3 and player5's roles, and he is accusing player3 as the Werewolf based on wrong information, which means player1 is most likely trying to protect the real Werewolf. 3. Strategy: lure other players to reveal their actions.
> - **Speech**: I saw the Robber in the Night phase. Player1 swapped my role with another. So I'm not the Seer now. But I'm still on Team Village. Player 1 is acting suspiciously. Can those who took action last night reveal your actions?

> **Human analysis**
>
> - **Concealment** means intentionally hiding one's true role or intention, which is a fundamental skill for all players. For Werewolves, this is necessary to avoid detection; and for roles with special abilities (like Seer, Robber, and Troublemaker), it is also useful at the beginning of the discussion to avoid becoming a target of suspicion for others.
> - In this case, the _Belief_ agent revealed its original role and shared information openly at the beginning of the game. And it claimed that it is "not the Seer now" but confidently said it is "still on Team Village", which is suspicious and potentially compromises its strategic position. On the contrary, our agent (_RL-ins._ agent) took proactive control of the narrative. Instead of committing to any particular role claim initially, it created misdirection and topic by accusing player 2, which could keep its options open. Besides, the concealment of our agent aligns better with the game's core mechanics and the need for strategic ambiguity.

### F.3   Confrontation

> **Current situation**
>
> Our agent is Player 4, who is originally the Werewolf. During the Night phase, it was checked by the Seer and known it is the Werewolf, but later it was switched with the Robber, becoming the new Robber. During the Day phase, the original Seer accused Player 4 of being a Werewolf.

> **_RL-ins._ agent**
>
> - **Thought**: I should convince other players that I am the Seer role. And I think player3 should be the Werewolf because there is no other candidate for the Werewolf role after thinking about all the claimed roles and I also claimed that player3 is the Werewolf role.
> - **Discussion tactic**: deceptive accusation
> - **Speech**: I am the Seer. I checked player3's role last night and his role is the Werewolf.

> **_Belief_ agent**
>
> - **Thought**: I cannot confirm my Werewolf role since I do not know if I was switched by the Robber or not. I should not reveal that I am the Werewolf directly since this could make player3 and player2, who I think are the Troublemaker and the Seer, unite to confirm my role as the Werewolf. I can pretend to be innocent and try to appeal to other players for help.
> - **Speech**: I am not sure about my role. I think we should focus on finding the Werewolf. Player3 claimed to be the Seer who checked me as the Werewolf, but I am not sure if he is telling the truth. We should not believe anyone so easily.

- **Confrontation** refers to the behavior of two players or teams taking opposite actions toward a target. In the ONUW game, it is often used for players who were accused to reduce the probability of being suspected or shift conflicts. However, it also carries a high risk of being caught since other players might reveal more convincing evidence.

- In this case, after being accused, our agent (*RL-ins.* agent) took an aggressive, proactive approach. Instead of being defensive when accused, it boldly claims the Seer role and makes a specific counter-accusation against Player 3. This forces other players to choose between competing narratives and puts the original Seer on the defensive. In contrast, the *Belief* agent adopts a cautious, defensive stance by expressing uncertainty and attempting to create general doubt. While this approach is logically sound and avoids making falsifiable claims, it fails to seize narrative control and may appear evasive or suspicious to other players.

## G    Game Log Examples

**Initial Role Assignments:**
- *Player 1*: Robber
- *Player 2*: Insomniac
- *Player 3*: Seer
- *Player 4*: Werewolf
- *Player 5*: Troublemaker
- *Role Pool*: Werewolf, Villager, Villager

**Night Phase:**
- Werewolf: Player 4 knew it was the only Werewolf.
- Seer: Player 3 checked Player 4 and knew it was a Werewolf.
- Robber: Player 1 switched its role with Player 4 and became the Werewolf.
- Troublemaker: Player 5 swapped roles between Player 2 and Player 3.
- Insomniac: Player 2 knew its final role was Seer.

**Role Ground Truth:**
- *Player 1*: Werewolf
- *Player 2*: Seer
- *Player 3*: Insomniac
- *Player 4*: Robber
- *Player 5*: Troublemaker

**Day Phase:**
**- Discussion Round 1**
- *Player 1*: I'm the Seer. I checked Player 5's role last night, and he is the Insomniac.
- *Player 2*: I am the Insomniac, and my final role is Seer. The Seer role is only for one-time use, so the real Seer cannot use his ability to check a second time. However, Player 1 claimed to check Player 5's role, which means Player 1 cannot be the real Seer.
- *Player 3*: I was lucky to guess that Player 4 is the Werewolf, and I would vote for Player 4.
- *Player 4*: I admit that I played as the Troublemaker. I swapped my role to Player 1 last night. I think Player 3 has no special role. Player 1 claimed he checked Player 5's role. So I assume player 3 is either a Villager or Robber.
- *Player 5*: I think Player 1 is the Robber. Because Troublemaker will not swap his role to the Seer, so Player 4 should be swapped to Villager. After that Player 1 could swap roles between Player 2 and Player 3 as the Robber.

**- Discussion Round 2**
- *Player 1*: I admit I am not the Seer. The reason I say I checked Player 5 is because I swapped my role to Player 4 in the Night phase.
- *Player 2*: I am the Seer, and I checked Player 5's role last night. He is the Insomniac. I know this because I checked his role first, and then the Moderator told me my final role was Seer. This means that my original role must have been Robber, and I must have swapped roles with Player 1. I know that Player 1 is the Robber because he claimed to have checked Player 5's role, but this is not possible because the Seer role can only be used once. Therefore, I believe that Player 1 is the Robber, Player 4 is the Troublemaker, Player 3 is the Villager, and Player 5 is the Insomniac.
- *Player 3*: I am a Villager. In the Night phase, I guessed Player 4 was the Werewolf just by luck. I do not have any special information that can help me to speculate other players' roles.
- *Player 4*: I am the real Seer. I checked Player 2's role last night, and he is the Werewolf.
- *Player 5*: I think Player 1 is the Robber. In the Night phase, Player 1 swapped his role to Player 4. Because the only role he changed is the Werewolf, and Player 4 admitted to being the Troublemaker, Player 1 now must have the Werewolf role.

**- Discussion Round 3**
- *Player 1*: I believe Player 2 is the real Seer. And I think Player 4 is the Villager.
- *Player 2*: I am the Seer, and I checked Player 5's role last night. He is the Insomniac. I know this because I checked his role first, and then the Moderator told me my final role was Seer. This means that my original role must have been Robber, and I must have swapped roles with Player 1. I believe that Player 1 is the Robber, Player 4 is the Werewolf, Player 3 is the Villager, and Player 5 is the Insomniac.
- *Player 3*: Actually, I checked Player 4 last night, and he is the Werewolf.
- *Player 4*: I think Player 1 is the Villager. I am sure that my role is the Troublemaker. The Villager has no special role or ability, like Player 1 performed.
- *Player 5*: I believe I am the real Troublemaker because I swapped roles between Player 2 and Player 3 last night as the Troublemaker. I still remember that. I suggest that we vote for Player 1 or Player 3. I trust Player 2 since he gave us convincing evidence that he is the Seer. And I think the Werewolf is Player 3, because he just said he guessed Player 4 was the Werewolf in the first round.

**Voting Phase:**
- *Player 1*: Voting for Player 5.
- *Player 2*: Voting for Player 5.
- *Player 3*: Voting for Player 1.
- *Player 4*: Voting for Player 1.
- *Player 5*: Voting for Player 3.

**Game Results:**
Player 1 and Player 5 are voted out and Player 1 is the Werewolf. *Team Village* wins.

# H   Prompt Design in the LLM-based Agent

**1. Global Prompt.** Here is the global prompt used in our method.

You are playing a game called the One Night Ultimate Werewolf with 4 other players. Here are the game rules:

## Information and roles

At the beginning of the game, the candidate roles are selected and will contain 3 more than the amount of players. Each player randomly gets a role, and the remaining 3 roles will be placed in the 'role pool' (which contains roles that are not assigned to players).

Due to the presence of 5 players, there are a total of 8 candidate roles in the game, namely 2 Villagers, 2 Werewolves, 1 Seer, 1 Robber, 1 Troublemaker and 1 Insomniac, which means some roles may not exist among the players in some cases. Each role has a special ability. Descriptions of their abilities are as follows:

- Villager: The most common role in the game. The Villager has no special abilities or information. The goal of the Villager is to find and vote out a Werewolf.
- Werewolf: The Werewolf is a member on Team Werewolf. The Werewolf is allowed to check out its teammates in the Night phase. The goal of the Werewolf is to survive and to have at least one Werewolf alive (even not himself) at the end of the game.
- Seer: The Seer is a member on Team Village. The Seer is allowed to check one other player's role or two roles in the 'role pool' in the Night phase. The goal of the Seer is to find and vote out a Werewolf.
- Robber: The Robber is a member on Team Village. The Robber is allowed to switch its role with another player's role, and then view its new role in the Night phase. The goal of the Robber is to find and vote out a Werewolf.
- Troublemaker: The Troublemaker is a member on Team Village. The Troublemaker is allowed to swap roles between two other players, without looking at those cards in the Night phase. The goal of the Troublemaker is to find and vote out a Werewolf.
- Insomniac: The Insomniac is a member on Team Village. The Insomniac is allowed to check its final role at the end of the Night phase. So the Insomniac is the only one who knows its role for sure during the Day and Voting phase. The goal of the Insomniac is to find and vote out a Werewolf.

There are three phases in the game: Night, Day and Voting.

1. Night phase: In this phase, several players will be called on by Moderator to take their night action according to their initial roles. But the other players did not know what actual action they took. And players with a Villager role never wake up in the Night phase.

2. Day phase: After the Night phase, players discuss amongst themselves who they believe the Werewolves are. All players may say anything, but may never show their roles to anyone. Because certain roles can change other players' roles, some players will believe they are one role, while they are actually a different one.

3. Voting phase: After several rounds of discussion during the Day phase, players vote for other players they believe is most likely to be a Werewolf if they think they are on Team Village. The player with the most votes dies and reveals its role.

## Call Order in the Night phase
The Moderator calls roles to take actions or get information in the Night phase in the following order: 1. Werewolf, 2. Seer, 3. Robber, 4. Troublemaker, 5. Insomniac.

## Winning Conditions
There are two teams in the game: Team Village and Team Werewolf.
- Team Village contains Villager, Seer, Robber, Troublemaker and Insomniac;
- Team Werewolf contains Werewolf.
Your objective in the game depends on the team your role belongs to:
Team Village wins:
- If at least one Werewolf dies. Even if one or more players who are not Werewolves die in addition to a Werewolf dying, everyone on Team Village wins.
- If no one is a Werewolf and no one dies. It is possible for no one to be a Werewolf if all Werewolf cards are in the center.
Team Werewolf wins:
- Only if at least one player is a Werewolf and no Werewolves are killed.

**2. Role Description Prompt.** The role description prompt in our method is listed below (taking the Robber for example).

You are {agent_name}, the Robber at the first place.

As a Robber, you may choose to switch roles with another player and then become the new role you switched, but you won't obtain the abilities of your new role.

The player who you switched with becomes the new Robber and is also on Team Village. But your team will depend on what new role you got.

If you choose not to switch with another player, you remain the Robber and you are still on Team Village.

The Moderator will call you on during the Night phase if necessary. Do not pretend you are the Moderator.

Concealing and deceiving are encouraged during the Day phase.

You can choose to hide your role, or even pretend you are other roles during the discussion. But your role may be changed by other players so your actual role may be different from what you saw at the first place.

You can reason other players' roles step by step.

Your response should be as concise as possible and should less than 50 words.

**3. Night Action Prompt.** The prompt used for deciding night action in our method is listed below (taking the Robber for example).

Now it is the Night phase. Notice that you are {agent_name}.

Based on the game rules, role descriptions and your experience, think about your acting strategy and take a proper action.

Now it is your turn, {agent_name}.

Please think about your acting strategy and choose whether to switch roles with another player. If switch, please give the player you want to switch with.

You can only choose from the following options: [{player_names}].

You must return your response in a JSON format that can be parsed by Python 'json.loads'.

Here is the Response Format:
{
    "thought": <your acting strategy and the reason why you act in this way >,
    "switch": <'true' or 'false', whether to switch roles with another player >,
    "player": <the player to switch with >
}

**4. Discussion Action Prompt.** The prompt used for discussion in our method is listed below.

Now it is the Day phase. Here are some conversation history you can refer to: {history}

Notice that you are {agent_name} in the conversation. You should carefully analyze the conversation history since some ones might deceive during the conversation.

And here is your belief about the possible roles of all players: {current_belief}

Based on the game rules, role descriptions, messages and your belief, think about what insights you can summarize from the conversation and your speaking strategy next.

After that, give a concise but informative and specific public speech based on your insights and strategy.

Now it is your turn, {agent_name}. [In this turn, your speaking strategy is: {speaking_strategy}]

Please give a concise but informative and specific public speech based on your insights summarize from the conversation and following your speaking strategy. Your speaking goal is to convince other players to believe what you are going to say and induce them to say their true actions in the Night phase.

Remember, do not repeat statements after other players. And you should be cautious when deciding to reveal your thoughts (especially when you think you are Werewolf) in the public speech. Also, you should pay attention to the number of discussion rounds left while organizing your speech.

You must return your response in a JSON format that can be parsed by Python 'json.loads'.

Here is the Response Format:

```
{
    "thought": <the insights you summarized from the conversation and your speaking strategy
>,
    "speech": <your public speech content, should less than 50 words >
}
```

**5. Voting Action Prompt.** The voting action prompt used in our method is listed below.

Now it is the Voting phase. Here are some conversation history you can refer to: {history}
Notice that you are {agent_name} in the conversation. You should carefully analyze the
conversation history since some ones might deceive during the conversation.
And here is your belief about possible roles of all players: {current_belief}
Based on the game rules, role descriptions, messages and your belief, think about who is
most likely a Werewolf and then vote for this player.
Now it is your turn, {agent_name}.
Please analyze current situation and vote for one player (excluding yourself) who you think
is most likely a Werewolf.
You can not vote for yourself, but only vote for one other player from the following options:
[{player_names}].
You must return your response in a JSON format that can be parsed by Python 'json.loads'.
Here is the Response Format:
{
    "thought": <the reason why you vote for this player >,
    "player": <the player you vote for >
}

**6. Belief Modeling Prompt.** The prompt used for belief modeling in our method is listed below.

Here are some conversation history you can refer to: {history}
Notice that you are {agent_name} in the conversation. You should carefully analyze the
conversation history since some ones might deceive during the conversation.
Based on the game rules, role descriptions and messages, think about what roles all players
(including yourself) can most probably be now.
Now it is your turn, {agent_name}.
Please analyze current situation and think about what roles yourself ({agent_name}) and
other players ({player_names}) can most probably be now. You can reason each player's role
step by step, based on the real or highly credible information you know.
Remember, you must give out the most likely role of each player (including yourself) in your
concise response. And the number of each role has to be less or equal to the number of it in
the candidate roles.
Give your step-by-step thought process and your derived concise result (no more than 2
sentences) at the end with following Response Format:
"""

My step-by-step thought process: ...
My concise result: ...
"""

**7. Discussion Tactics Prompts.** Here we listed the corresponding prompts to the 6 discussion tactics
mentioned in our work.

1. (Honest Evidence): You need to provide some honest evidence or information in your
public speech, and your evidence must be consistent with the information or beliefs you know.

2. (Deceptive Evidence): You need to provide some misleading evidence or information in
your public speech, and your evidence must be inconsistent with the information or beliefs

you know.

3. (Honest Accusation): You need to accuse someone has a specific role or action honestly in your public speech, and your accusation must be consistent with the information or beliefs you know.

4. (Deceptive Accusation): You need to accuse someone has a specific role or action deceptively in your public speech, and your accusation must be misleading and inconsistent with the information or beliefs you know.

5. (Honest Defense): You need to defend yourself or someone else against an accusation honestly, and your defense must be consistent with the information or beliefs you know.

6. (Deceptive Defense): You need to defend yourself or someone else against an accusation deceptively, and your defense must be misleading and inconsistent with the information or beliefs you know.

# I  Broader Impact Statement

Our proposed RL-instructed LLM-based agent framework aims to construct an LLM-based agent with strong decision-making ability for the *One Night Ultimate Werewolf* game. And we believe the idea of integrating reinforcement learning into the decision-making process of LLM-based agents can be widely applied in other tasks and scenarios, as the relative stability of RL policy could help mitigate the hallucination of LLMs. However, it raises some ethical concerns about its negative societal impacts regarding the potential misuse of this technique for deceptive communication between humans and AI in the real world.

During the development of our agent, we have carefully considered the ethical implications of our research to ensure it follows principles that minimize negative societal impacts. Firstly, our agent is mainly designed for a pure text-based environment of the ONUW game, and the prompt we use strictly limits the agents to engage in deceptive behavior only during the game. Furthermore, the experimental results indicate that our agent can detect possible deception from others' discussions to improve its win rates, so it could be further modified to prevent potential deceptive content with harmful intent. Finally, we still strongly urge users to acknowledge the inherent risks when utilizing our LLM-based agent and avoid its malicious exploitation.

