# OpenReview forum: "Learning to Discuss Strategically: A Case Study on One Night Ultimate Werewolf"
_NeurIPS.cc/2024/Conference — NeurIPS 2024 poster_

### Official Review · Reviewer_b5Fq · 2024-07-01

**Soundness:** 4
**Presentation:** 3
**Contribution:** 3
**Rating:** 7
**Confidence:** 3

**Summary:**

The authors take a specific natural language game (One Night Ultimate Werewolf) and study the performance of RL-trained LLM agents with respect to the mathematically derived (by the authors) Nash equilibrium solutions.

**Strengths:**

[1] The paper chooses ONUW as a game that has the Nash equilibrium solution to be used as an oracle/reference. The paper then applies learning to improve the 0-shot answers/behavior of LLMs. The learning is performed with offline RL over the collected game episodes.

[2] Even though the paper laser-focuses on ONUW, its methodology and results can be a valuable stepping stone to solving real-life problems.

[3] As negotiations are an integral part of the society, its automation and improvement of effectiveness is a valuable direction of research. This paper makes a small but definite step in the direction of automated LLM-powered negotiations.

[4] The authors do a great job of rigorous and comprehensive research of ONUW optimal strategies for players by proving theorems for several specific cases.

[5] The formalized way to assess the performance of the learned policies with the help of NashConv (distance to equilibria) is very valuable and insightful.

[6] In 6.3 and 6.4 the authors show that the policy learned with ChatGPT4 can be transferred to Gemini thus demonstrating the generalization across backend LLMs.

[7] As described in E.2 the use of text embeddings to encode the state is intriguing.

[8] The broader impact is well discussed.

**Weaknesses:**

[1] Per se the entire paper is bound to the ONUW game and its rules. The ability of the bundle of proposed algorithms and proofs to transfer to different games or game-like environments like financial markets is not demonstrated.

[2] I have not found any examples of the discussions between the LLM players in the manuscript. Further, the code is not provided by the authors. Thus there is no way to see the actual discussions between the LLM players and assess the value and quality of the generated discussion.

[3] Artificial prior knowledge is inserted into the solution that constraints the model to choose between 6 options: Honest/Deceptive and Evidence/Accusation/Defense. The proposed model is seemingly limited by these options and does not allow learning or discovery of alternative discussion strategies.

[4] It would be good to have a random choice baseline and compare the proposed method to its performance.

[5] The paper studies a game with one round of negotiations that has limited practical use in the light of the fact that negotiations normally are conducted in several rounds.

**Questions:**

[1] What was the reasoning behind choosing offline RL over online RL?

[2] How is the classification into Honest/Deceptive and Evidence/Accusation/Defense performed: manually or by an 0-shot LLM?

[3] In “An ε-Nash equilibrium is found under the
74 limitation of both human-side and werewolf-side strategies [28]
” maybe villager-side?

[4] As per Table 3 state dim is 1536 which seems quite high for ONUW. What is the reasoning behind selection of this specific number? Is there any graph of a sweep across the state dimension as a hyperparameter.

[5] In E.4 the USD cost per game is discussed. What is the total cost of the experiments described in the paper?

**Limitations:**

The limitations are well discussed or clear in principle.

---

> ### Author Rebuttal · Authors · 2024-08-07
>
> Thank you very much for your positive comments.
>
> **Response to W1: The transferability of proposed algorithms and proofs to different games or game-like environments like financial markets is not demonstrated.**
>
> Thank you for your comment. As this work is initially motivated by our analyses of the ONUW game, we mainly focus on the ONUW game in the experiments. We recognize that the financial markets is an interesting and highly practical environment for demonstrating our agents, but it needs relevant knowledge to construct agents for this scenario. However, since the agent framework we proposed is modular, we believe it can be transferred to various scenarios with similar discussion-driven dynamics like bargaining in the financial markets, as long as we have certain knowledge or data to train a corresponding discussion policy.
>
> **Response to W2: I have not found any examples of the discussions between the LLM players in the manuscript. Further, the code is not provided by the authors.**
>
> Thank you for your constructive comment. We made further human analyses based on the discussion logs between LLM players. However, due to the page limits in the global rebuttal, we can only provide one analysis in the PDF file. More analysis results and discussion examples will be updated in the revision. Also, the anonymized link to our code has been sent to the Area Chair in a separate comment as required by the conference guidelines.
>
> **Response to W3: The proposed model is seemingly limited by the predefined discussion tactics and does not allow learning or discovery of alternative discussion tactics.**
>
> Thank you for your comment. As discussed in our work, the predefined six discussion tactics were intended to simplify the problem and facilitate the learning process, due to the lack of a standardized classification for discussion tactics and adequate human player data in games like ONUW. While this approach does constrain the agents to choose from a predefined set of tactics, it also provides a structured framework for understanding the dynamics of discussion. However, as part of future work, we consider it interesting to explore more flexible and adaptive methods that allow for the discovery of new discussion tactics as well.
>
> **Response to W4: It would be good to have a random choice baseline and compare the proposed method to its performance.**
>
> Thank you for your constructive comment. We have added an LLM-based agent whose discussion policy is to randomly choose discussion tactics as a new baseline (named *Random*). Please refer to the global response for the results.
>
> **Response to W5: About the number of negotiation rounds.**
>
> Thank you for your feedback. Actually, although the ONUW game has only one Night, one Day, and one Voting phase, there could be many rounds of discussions during the Day phase where players are allowed to discuss, depending on the game settings. And in our settings, we allowed players to have three rounds of discussions. Please refer to the PDF file in our global response for the detailed game process. Moreover, we consider our discussion policy can adapt to multiple rounds of discussions since it relies on the current state of discussion rather than a specific number of rounds.
>
> **Response to Q1: The reason for choosing offline RL.**
>
> Thank you for your comment. The main reason we chose offline RL is the slow interaction with LLMs, especially the interaction rate limits of GPT-4. Online RL requires agents to interact with the environment in real-time to collect data for training. However, as some parts of our agent framework are LLMs and each inference of LLMs takes a few seconds, it is almost impossible for our agents to train the discussion policy in an online manner. So we decide to collect game data first and then adopt offline RL for training.
>
> **Response to Q2: How is the classification into six discussion tactics performed: manually or by an 0-shot LLM?**
>
> The classification of players' discussion tactics is performed by a 0-shot LLM.
>
> **Response to Q3: About the "human-side" or "villager-side".**
>
> Thank you for pointing it out. The "human-side" is the name that reference [28] used to represent the "villager-side", oppositing to the "werewolf-side". As it might cause confusion for readers, we have replaced it with "villager-side" in our latest version to keep consistency with other parts of the paper.
>
> **Response to Q4: What is the reasoning behind selection of this specific number 1536 for state embeddings?**
>
> Thank you for your insightful comment. The 1536 state dim is actually decided by the model we selected for state embeddings. In our experiments, GPT's `text-embedding-ada-002` model is adopted to encode players' observations and beliefs into state embeddings. We believe it is well pretrained and could effectively represent the semantic information of the original text. We have not performed a hyperparameter sweep across different state dims as it is not the main scope of our work, but it is an interesting suggestion for future work to explore the impact of varying the state dim.
>
> **Response to Q5: What is the total cost of the experiments described in the paper?**
>
> Thank you for your comment. For each evaluation in our experiments, we repeated the game 30 times to obtain the final results. Since we conducted experiments on one three-player game (Sec 6.2), two five-player games in fixed settings (Sec 6.3), and one five-player game in a random setting (Sec 6.4), there are 50 sets of results. However, the cost of each agent version (such as *ReAct* and *LLM-instructed*) varies a lot due to their frameworks, and the Gemini API was free when we conducted our experiments, so we can only estimate the total cost be around 200.

---

> > ### Comment · Reviewer_b5Fq · 2024-08-11
> > **Response to the rebuttal**
> >
> > Thank you for your response. As a side note, it is interesting to see ReAct performing worse than random for GPT4. I am keeping my score.

---

> > > ### Author Response · Authors · 2024-08-12
> > > **Thank you for your reply!**
> > >
> > > Thank you for your reply! And yes, judging by the win rates, the *Random* agent performs better than the *ReAct* agent for GPT-4. But it is also notable that the average number of votes got by the *Random* agent is greater than that of the *ReAct* agent. We believe that the performance between the *Random* and *ReAct* agents is actually similar, and the slightly higher win rate of the GPT-4-based *Random* agent might owe to their teammates during the gameplay.

---

### Official Review · Reviewer_Tkt3 · 2024-07-12

**Soundness:** 3
**Presentation:** 3
**Contribution:** 4
**Rating:** 8
**Confidence:** 4

**Summary:**

This paper delves into the strategic aspects of discussion in the context of a population social deduction game, "One Night Ultimate Werewolf" (ONUW) game. By analyzing Perfect Bayesian Equilibria in scenarios with and without discussion, the authors highlights the pivotal role of discussion tactics in influencing players' beliefs and utilities. They propose a novel framework that employs reinforcement learning to instruct an LLM agent, enabling it to decide and execute strategic discussion policies. The authors empirically show that their framework can recognize and approximate the equilibira, and achieve strong performance across diverse ONUW settings.

**Strengths:**

1. The paper introduces a new complex communication game, ONUW, and provides a clear explanation of the problem formulation. The ONUW game offers interesting role deduction challenges and seems it can be a benchmark for evaluating the deduction ability of LLM agents.
2. The authors present thoroughly theoretical analyses on a specific setting of the ONUW game, which prove the significance of discussion on influencing other players' belief and provide a solid foundation for their following proposed method.
3. The idea of integrating RL policy into LLM agents to improve their strategic ability in communication games is innovative. It has the potential to be applied in future work related to strategic LLM agents in various situations. And the empirical results shows the efficacy of this method.

**Weaknesses:**

1. By integrating RL policy into the reasoning process, the authors aim to improve the discussion ability of LLM agents, which provides a new angle for constructing LLM agents. However, applying RL policy to select discussion tactic seems analogous to selecting action candidates in related work [1].
2. Since the RL policy is trained on game logs generated by LLM agents, it is possible that players in the training dataset appear again during testing, resulting in less convincing results.
3. Typos：
   1. In line 115, 126, player i's information state should be $h^i \in \mathcal{H}^i$.
   2. The symbol $q$ is used twice as the probability that Werewolves vote for Player 3 (line 194), and as the discussion policy in Equation 6.
   3. In line 224-225, it seems that $\theta^i$ refers to player i's derived belief on all players' types, while $\theta$ refers to the groundtruth. These two symbols should be further distinguished.

> [1] Zelai Xu, et al. "Language agents with reinforcement learning for strategic play in the werewolf game".

**Questions:**

1. It is a little confusing why the dimension of Player 2's belief $b^2$ is 6 while the dimension of Player 1's belief $b^1$ is 3 in the formal Theorem D.2.
2. Can the authors give a detailed explanation of why the NashConv values of GPT-4-based agents are higher than Gemini-based agents?

**Limitations:**

The authors have stated the limitations about their work in Section 7.

---

> ### Author Rebuttal · Authors · 2024-08-07
>
> Thank you very much for your insightful review.
>
> **Response to W1: Differences from related work [1].**
>
> Thank you for your comment. We have discussed the differences between our work and related work [1] (line 80-82). Here are more detailed differences. And these will be added to the main paper in a future version.
>
> The main differences between our work and [1] lie in the model architecture and the training process. In related work [1]'s agent model, a set of action candidates is generated and the RL policy is trained to select one action among these candidates. However, the RL policy in our framework (*i.e.* discussion policy) is directly integrated into the thinking and decision-making process of agents. So the RL policy in [1] can be seen as an adjustment for the action distribution of the LLM-based agent, while our discussion policy is part of the agent's strategy. As for the training process, related work [1] adopts a population-based RL training method to adapt to the diverse actions of various players. We believe the discussion policy should be generic and invariant across players, so it is trained in a single-agent manner. Finally, the experimental results of our agents in both three-player and five-player games indicate good scalability, which is not explicitly shown in [1].
>
> **Response to W2: It is possible that players in the training dataset appear again during testing, resulting in less convincing results.**
>
> Thank you for your feedback. In fact, most agents in our experiments are different from those used for generating game logs for training. Specifically, the players in the training dataset only utilize the GPT-4-based LLM-instructed agent. On the contrary, the players in our experiments have various settings, including *ReAct*, *Belief*, *LLM-instructed*, and *RL-instructed*, while adopting Gemini or GPT-4 as the backend LLM in different experiments. For example, all players in Figure 4 use Gemini as the backend LLM to avoid the potential appearance of players in the training dataset.
>
> We believe the results are convincing since these game logs only reflect the preferences of GPT-4 which are supposed to be different from Gemini. Also, the differences between agents' versions have an impact on players' performances as shown in our experimental results, so we can regard them as different players.
>
> **Response to W3: About typos.**
>
> Thank you for pointing out the mistakes. We have fixed these typos in our revision.
>
> **Response to Q1: About the dimensions of players' beliefs.**
>
> We appreciate the reviewer for pointing it out, as it might confuse readers who are not familiar with game theory. Each belief in our theorem corresponds to a probability distribution over a specific information set. Therefore, the dimension of belief is actually the number of states in its corresponding information set. According to the game rules, players should vote for other players simultaneously in the Voting phase, which can be seen as a normal-form subgame. However, the game tree is used to describe the extensive-form game, so we need to transform the Voting phase into the extensive form for better demonstration:
>
> - In our game tree (Fig 2 and 5), we assume Player 2 votes after Player 1 and Player 3 votes after Player 2.
> - For Player 1, since it does not know which action Player 3 (original Robber) takes at night, there are 3 potential states in its information set.
> - For Player 2, it does not know Player 1's voting choice as they vote simultaneously in fact, and it also does not know Player 3's action at night, so there are 6 potential states in its information set in the game tree considering Player 1 votes before it.
>
> Therefore, the different belief dimensions are actually due to the manually set "pseudo" voting order. But it has no impact on the final derived equilibria results.
>
> **Response to Q2: Detailed explanation of why the NashConv values of GPT-4-based agents are higher than Gemini-based agents.**
>
> The NashConv value is defined as $\text{NashConv}(\pi)=\sum_i\left[R^i(\text{BR}(\pi^{-i}), \pi^{-i}) - R^i(\pi)\right]$, which represents how much utilities players can gain by deviating to their best responses. It is notable that players' gains in NashConv actually relate to other players' strategies $\pi^{-i}$, so the NashConv value is a result of all players' strategies.
>
> Let us still take the three-player game as an example. In this game, Player 1's action space is $A^1=\{V2, V3\}$, Player 2's action space is $A^2=\{V1, V3\}$, and Player 3's action space is $A^3=\{(NS, V1), (NS, V2), (S1, V1), (S2, V2)\}$. We ignore the $(S1, V2)$ and $(S2, V1)$ in $A^3$, since they are respectively dominated by $(S1, V1)$ and $(S2, V2)$ (this result is stated in Appendix D.1). Considering the following two strategy profiles:
>
> 1. $\pi_1$: Player 1 and Player 2 always vote for Player 3, while Player 3 adopts a random strategy. In this case, $\text{NashConv}(\pi_1) = 1/2+1/2+0 = 1$.
> 2. $\pi_2$: All players adopt random strategies. And in this case, $\text{NashConv}(\pi_2) = 1/2+1/2+1/4=5/4$.
>
> We can see that even though the deviation gains of Player 1 and Player 2 are the same in $\pi_1$ and $\pi_2$, the deterministic strategies of Player 1 and Player 2 in $\pi_1$ make Player 3 can no longer find a better response, resulting the lower NashConv value comparing to $\pi_2$.
>
> When we analyze the evaluation logs of the experiments in Section 6.2, we find that when employing Gemini, the two Werewolves (Player 1 and Player 2) are more likely to both vote for Player 3, while the GPT-4-based agents are more random. It suggests that the Gemini-based agents' strategy profile is closer to $\pi_1$, so its NashConv value is lower than the GPT-4-based agents in the same settings.
>
> **References:**
>
> [1] Zelai Xu, et al. Language agents with reinforcement learning for strategic play in the werewolf game. arXiv preprint arXiv:2310.18940, 2023.

---

> > ### Comment · Reviewer_Tkt3 · 2024-08-10
> >
> > Thanks for your detailed response. Most of my concerns have been addressed. I will raise the score.

---

> > > ### Author Response · Authors · 2024-08-10
> > >
> > > Thank you for your valuable feedback and for adjusting your evaluation of our work. We appreciate your acknowledgment of our efforts.

---

### Official Review · Reviewer_ADZR · 2024-07-14

**Soundness:** 3
**Presentation:** 3
**Contribution:** 2
**Rating:** 6
**Confidence:** 3

**Summary:**

The authors investigate the social game ‘One Night Ultimate Werewolf’ (ONUW) as a testbed for their framework on RL-instruct-tuning an agent to select optimal discussion tactics. They prove the existence of a Perfect Bayesian Equilibria in the game ‘Werewolf’ when the game consists of a single round and show the effectiveness of their approach in a three-player setting as well as a five-player setting to prove generality.

**Strengths:**

The authors introduce a novel RL-instructed LLM-based agent framework specifically tailored for strategic discussions in ONUW focused on identifying the best discussion tactics in communication scenarios as well as providing an environment for further testing.

In addition, the authors further provide a theoretical analysis of the ONUW game, formulating it as a Multi-Phase Extensive-Form Bayesian Game and establishing the existence of Perfect Bayesian Equilibria in different scenarios

**Weaknesses:**

The primary weakness of the paper is noted by the authors, where the discretization of the space of discussion tactics results in what could be an oversimplification of the dynamics that would typically occur in such a game. For example, what could occur in a typical werewolf game is that the werewolf makes no effort to defend themselves, implying the other villagers are bullying them in an attempt to garner false sympathy.

The authors also acknowledge another weakness in that identifying and discretizing the space of discussion tactics in a game is a manual and time-consuming process. While the experiments in the paper imply generalization to X number of players, this generalization is still limited to the ONUW game.

A final weakness is that the results are somewhat limited by the experiments only being performed with LLMs rather than including human players as well. However, the reviewer acknowledges the additional time and monetary costs performing such experiments would occur.

**Questions:**

How did the authors determine the specific categories for discussion tactics, and have they considered any automated methods for discovering these categories?

Did the authors consider including any human participants in their trials? If not, why?

How do the authors imagine the role-switching dynamics of ONUW impacted their results? What might change were the roles fixed?

**Limitations:**

The authors touch on some limitations of their work, however it is unclear whether the topics addressed in the questions (the impact of human participants, the word required to pre-define the decision tactics) were not included due to being outside the scope of the work, or not relevant given some metric or detail the reviewer may have missed.

---

> ### Author Rebuttal · Authors · 2024-08-07
>
> Thank you very much for your positive feedback.
>
> **Response to W1: About the oversimplification and manual discretization of the discussion tactic space.**
>
> Thank you for highlighting this limitation. As acknowledged in our work, the six discussion tactics are manually identified and simplified. This is primarily due to the lack of a standardized classification for discussion tactics and adequate human player data in games like ONUW. But the manual discretization of discussion tactic space provides a structured framework for understanding the dynamics of discussion during the game, and the experimental results have to some extent demonstrated the effectiveness of learning a discussion policy based on these tactics to improve the discussion ability of LLM-based agents. And as part of future work, we consider it interesting to explore techniques to automatically identify the discussion tactics during the gameplay as well.
>
> **Response to W2: The generalization is still limited to the ONUW game.**
>
> Thank you for your comment. Since the manually selected discussion tactics are specific to the ONUW game, the trained discussion policy is naturally limited to this game. However, the framework we proposed is modular and can be adapted to various scenarios with similar discussion-driven dynamics. For example, if we can somehow collect widely used bargaining tactics, then we could train a bargaining policy in the same way and apply it to our framework to construct an LLM-based agent for bargaining.
>
> **Response to Q1: How did the authors determine the specific categories for discussion tactics?**
>
> Thank you for your comment. The specific categories for discussion tactics were determined mainly through the inspiration from prior research on argumentation [1, 2] and analyzing human choices when playing similar games. And to highlight the potential deception in the ONUW game, we further divide these tactics into *honest* and *deceptive* ones. We have not yet explored automated methods for discovering these categories, but we highly acknowledge it is an interesting avenue for future research, as it could enhance the automation level of the LLM-based agent constructions and might discover unexpected tactics among humans' discussions.
>
> **Response to Q2&W3: Did the authors consider including any human participants in their trials?**
>
> Yes, we consider it would be interesting to include human players to play with the agents as well. As the main topic of our work is to demonstrate the importance of discussion in the ONUW games and to improve the discussion ability of LLM-based agents, we did not consider the impact of human participants in our experiments. However, we are planning to add a human interaction interface to our project code, which will be updated soon.
>
> **Response to Q3: How do the authors imagine the role-switching dynamics of ONUW impacted their results? What might change were the roles fixed?**
>
> Thank you for your insightful comment. As we analyzed in Section 4 and the game tree, the role-switching dynamics of ONUW do significantly impact the game's outcome, players' utilities, and the strategies employed by players. And it is the role-switching dynamics that make the three-player ONUW game possible. Imagine if roles were fixed in the three-player case, then two Werewolves would know the player left must be on *Team Village* and vote it out together, which would make the game meaningless.
>
> Also, we believe that the reasoning difficulty brought by the role-switching dynamics is one of the key challenges that reflect the abilities of different language models. If roles are fixed, the ability gaps between different models in the experimental results would be narrowed down, and the need for strategic deception and counter-strategies would be reduced, possibly leading to simpler discussion policies.
>
> **References:**
>
> [1] Bolin Lai, et al. Werewolf among us: A multimodal dataset for modeling persuasion behaviors in social deduction games. arXiv preprint arXiv:2212.08279, 2022.
>
> [2] Winston Carlile, et al. Give me more feedback: Annotating argument persuasiveness and related attributes in student essays. In Proceedings of the 56th Annual Meeting of the Association for Computational Linguistics (Volume 1: Long Papers), pages 621–631, 2018.

---

> > ### Comment · Reviewer_ADZR · 2024-08-13
> >
> > Thank you for your explanations. I will keep my current score.

---

> > > ### Author Response · Authors · 2024-08-13
> > >
> > > Thank you for your reply! We sincerely appreciate your valuable reviews and acknowledgment of our efforts.

---

### Official Review · Reviewer_uoU5 · 2024-07-15

**Soundness:** 3
**Presentation:** 3
**Contribution:** 2
**Rating:** 4
**Confidence:** 4

**Summary:**

The paper presents an innovative framework for enhancing the discussion capabilities of language agents in the game "One Night Ultimate Werewolf" (ONUW) using reinforcement learning (RL). The authors propose a multi-phase extensive-form Bayesian game formulation for ONUW, analyze perfect Bayesian equilibria in both discussion and non-discussion scenarios, and develop an RL-trained discussion policy. Experimental results demonstrate the effectiveness and generalizability of the framework across various game settings.

**Strengths:**

1. The theoretical analysis of the game, including the formulation as a Bayesian game and the derivation of equilibria, is thorough and well-executed. The experimental design is robust, utilizing state-of-the-art LLMs and a novel RL training methodology.

2. The paper is well-structured and clearly written. The authors provide detailed explanations of the game mechanics, the theoretical framework, and the RL training process, which make the complex content accessible to readers.

**Weaknesses:**

1. Comparison with Related Work: While the paper provides an innovative approach to using RL and LLMs in the Werewolf game, there is a need for a more detailed comparison with closely related works, particularly those combining LLMs with reinforcement learning strategies. For instance:
Xu et al. (2023) [1] also explore strategic play in the Werewolf game using language agents trained with reinforcement learning. A comparative analysis highlighting what differentiates the current approach from Xu et al.'s methodology would clarify the novelty and the specific advancements made.
Wu et al. (2024) [2] utilize offline RL and a dataset-driven approach to enhance reasoning in LLMs within the same game context. Discussing how the methodologies differ, especially in terms of model training, dataset utilization, and resultant agent behavior, would strengthen the current work's positioning within the field.

2. Lack of Human Evaluation: The experimental section primarily focuses on win rates to demonstrate the effectiveness of the proposed framework. However, Werewolf (ONUW) involves complex human interactions and strategic discussions that might not be fully captured by win rates alone. The game's social and psychological aspects, such as bluffing and persuasion, are crucial:
It would be beneficial to include human evaluations to assess the quality of the AI's gameplay and its ability to mimic human-like strategic discussions. This could involve subjective assessments from experienced human players regarding the AI's ability to integrate seamlessly into human gameplay, its strategic depth, and its communication effectiveness.

LLM+RL：[1] Zelai Xu, Chao Yu, Fei Fang, Yu Wang, and Yi Wu. Language agents with reinforcement learning for strategic play in the werewolf game. arXiv preprint arXiv:2310.18940, 2023.
LLM+offline RL, dataset: [2] S Wu, L Zhu, T Yang, S Xu, Q Fu, Y Wei, H Fu, Enhance reasoning for large language models in the game werewolf.  arXiv preprint arXiv:2402.02330, 2024

3. Dataset Size and Composition: Detailed information on the size and composition of the dataset used for offline RL is necessary. Understanding the diversity and representativeness of the game logs in the dataset would help in assessing the potential generalizability and robustness of the trained models. This includes the number of game sessions, variety of player strategies, and the range of game outcomes included.

**Questions:**

1. Detailed Comparison with [1] and [2]: Can the authors provide a more detailed comparison of their work with the approaches in [1] Zelai Xu et al., 2023, and [2] S Wu et al., 2024? Specifically, how does the integration of RL and LLMs in your framework differ from these studies in terms of model architecture, training processes, and performance metrics?

2. Human-Like Gameplay Evaluation: Given the social and psychological complexities of the ONUW game, how do the authors plan to evaluate the AI's performance in terms of human-like behavior and strategic discussion quality? Are there plans to incorporate human player evaluations, and if so, what methodologies would be used to assess the AI's gameplay against human strategies and interactions?

3. The authors note "we additionally contribute a dataset featuring players employing various discussion tactics in the ONUW
game. ". Where is the dataset, can the authors opensouce the dataset?

4. Clarification on Offline RL Implementation: The paper mentions the use of offline RL due to the slow interaction with LLMs, but it lacks specific details on how the LLM embeddings are handled within the offline RL framework. Are the state embeddings used by the discussion policy generated by running LLMs on the offline dataset prior to training? If so, how is the freshness and relevance of these embeddings ensured over iterations of RL training?

**Limitations:**

The authors should consider expanding the discussion on the limitations related to the discretization of discussion tactics and the potential over-reliance on specific datasets. Suggestions for future work could include exploring methods for dynamic tactic generation or adjustment based on real-time gameplay feedback, which could help in developing more adaptable and robust AI agents for complex communication games. Additionally, addressing the computational demands and proposing more resource-efficient models could make the technology more accessible for broader applications.

---

> ### Author Rebuttal · Authors · 2024-08-07
>
> Thank you very much for your valuable feedback.
>
> **Response to Q1&W1: Detailed comparison with related work [1] and [2].**
>
> Thank you for your constructive comment. We have discussed the differences between our work and related work [1] (line 80-82). Here are detailed differences between our work with related works. And these will be added to the main paper in a future version.
>
> - **Differences from Xu et al. (2023) [1]:** The main differences lie in the model architecture and the training process. In related work [1]'s agent model, a set of action candidates is generated and the RL policy is trained to select one action among these candidates. However, the RL policy in our framework (*i.e.* discussion policy) is directly integrated into the thinking and decision-making process of agents. So the RL policy in [1] can be seen as an adjustment for the action distribution of the LLM-based agent, while our discussion policy is part of the agent's strategy. As for the training process, related work [1] adopts population-based RL training method to adapt to the diverse actions of various players. We believe the discussion policy should be generic and invariant across players, so it is trained in a single-agent manner. Finally, the experimental results of our agents in both three-player and five-player games indicate good scalability, which is not explicitly shown in [1].
> - **Differences from Wu et al. (2024) [2]:** From the perspective of the model architecture and usage, RL in related work [2] is utilized to train the *Thinker* module, which focuses on handling complex logical analysis and strategic planning in specialized tasks, while the RL policy in our work (*i.e.*, the discussion policy) mainly focuses on enhancing the discussion abilities of LLM-based agents. In the training process, the input and output of the RL policy in [2] are structured language features. But the input in our discussion policy is the embedding of current observation and belief and the output is the chosen discussion tactic. Meanwhile, as the context generated by the *Presenter* in [2] is supposed to be consistent with *Thinker*'s produced policies, the agent framework in [2] barely considers the significance of speaking strategically (*i.e.* being honest or deceptive), which is actually the motivation and core of our work.
>
> **Response to Q2&W2: The human evaluation for agents' performance.**
>
> Thank you for your comment. We understand the importance of human evaluations in assessing the quality of AI's performance and its ability to mimic human-like strategy discussions. We have conducted several human evaluations and analyses based on the game logs. However, due to the page limits in the global rebuttal, we can only provide one typical analysis in the PDF file. More results will be added to the appendix in the revision.
>
> **Response to W3: Detailed information of the dataset.**
>
> Thank you for your comment. We have provided the process of data collection, the dataset statistics and analysis in Appendix E.1. The dataset used for offline RL consists of 120 game logs (containing 1800 discussion turns) from the five-player ONUW game. And for each log, it includes the entire discussion history, players' discussion tactics, the initial and final role assignment, voting results, and game outcomes.
>
> **Response to Q3: Can the authors opensource the dataset?**
>
> Thank you for your feedback. We highly acknowledge the importance of opensourcing the code and dataset of our work. The anonymized link to our code has been sent to the Area Chair in accordance with the conference guidelines. And right now we are cleaning our dataset, which is scheduled to be opensourced after the rebuttal period.
>
> **Response to Q4: Clarification on offline RL implementation.**
>
> Thank you for your comment. Our offline RL training details can be found in Appendix E.2. In our implementation, we first use GPT's `text-embedding-ada-002` model to encode players' observations in the dataset into state embeddings, and then adopt these embeddings for further offline RL training. The reason we adopted a frozen encoder is that GPT's embedding model is well pretrained and we believe it could effectively represent the information of the original text in the semantic space. Also, related work [1] adopts prior embeddings before training. However, we agree with the reviewer's point that fine-tuning the embedding model may help further improve the performance of the agents. We will clarify this in the revision.
>
> **References:**
>
> [1] Zelai Xu, et al. Language agents with reinforcement learning for strategic play in the werewolf game. arXiv preprint arXiv:2310.18940, 2023.
>
> [2] Shuang Wu, et al. Enhance reasoning for large language models in the game werewolf. arXiv preprint arXiv:2402.02330, 2024.

---

> > ### Comment · Reviewer_uoU5 · 2024-08-13
> >
> > Thank you for the response! After considering other reviews and responses, I've decided to maintain my rating.

---

> > > ### Author Response · Authors · 2024-08-13
> > >
> > > Thank you for your consideration. Please let us know if you have any further questions or concerns. We will respond to your inquiries as promptly as possible. If we can hopefully address your concerns, we hope you'll consider adjusting your rating.

---

### Author Rebuttal · Authors · 2024-08-07

We thank all reviewers for their thoughtful feedback! Your reviews have greatly improved the paper. And we are grateful for the appreciation of our theoretical analysis (`uoU5`, `ADZR`, `Tkt3`, `b5Fq`), the innovation of our framework (`ADZR`, `Tkt3`, `b5Fq`), and the recognition of our research scenario (`Tkt3`, `b5Fq`).

We have responded to each reviewer's respective questions below. Furthermore, we will also revise the paper accordingly to address all other suggestions and comments. For the code of our project, we have forwarded an anonymous GitHub repository to the Area Chair in accordance with the conference guidelines. It contains the implementation of the ONUW game environment, all agents used in the experiments, and the dataset process and policy training procedure. We hope our responses could clarify existing questions.

**Response to Reviewer b5Fq's suggestion on adding a random choice baseline.**

We added an LLM-based agent whose discussion policy is to randomly choose discussion tactics as a new baseline (named *Random*), and conducted experiments in the same settings as other agents in Table 1. Here are the results:

| Agents | Gemini (*Belief*) Win Rates | Gemini (*Belief*) Avg. Votes | GPT-4 (*Belief*) Win Rates | GPT-4 (*Belief*) Avg. Votes |
|-------|--------|--------|--------|--------|
| *ReAct*  | 0.40 | 1.23 | 0.30 | 1.73 |
| *Belief* | 0.40 | 1.73 | 0.32 | 1.87 |
| *Random* | 0.37 | 1.53 | 0.32 | 2.03 |
| *LLM-ins.* | 0.62 | 1.10 | 0.37 | 1.90 |
| *RL-ins.* | **0.70** | 1.10 | **0.50** | 1.87 |

It can be seen that the performance of the *Random* agent is almost similar to that of the *ReAct* and *Belief* agents, demonstrating the significance of strategic discussion policy. Especially, when playing with GPT-4-based *Belief* agents as Player 3, the *Random* agent is more likely to be voted. We recognize it is because the random discussion policy increases the inconsistency in agent's discussions.

**About the PDF file.**

In the supplementary PDF file, we provided a graph about the detailed game process of the ONUW game, and a typical example of the agents' performance along with human analysis.

---

### Author Response · Authors · 2024-08-12

Dear AC and Reviewer uoU5,

As the author-reviewer discussion period is drawing to a close, we kindly request your attention to our rebuttal. We hope we have addressed the concerns raised, particularly regarding (1) open-sourcing, for which we've provided an anonymized code link to the AC as per conference guidelines; (2) human player evaluation, where we've included a typical analysis in the attached PDF and committed to adding more results in the appendix of our revised version; and (3 ) detailed comparisons with related work [1] and [2], which we've elaborated on in our response and will incorporate into the main paper in a future version.

Please let us know if you have further questions and we're more than willing to discuss them.

Best regards,
Authors of Submission 9801

---

### Decision · Program_Chairs · 2024-09-25

**Decision:**

Accept (poster)

**Comment:**

This paper studies the impact of strategic discussion in the game One Night Ultimate Werewolf (ONUW).  It gives an equilibrium analysis of the game with and without discussion, implements an LLM-based RL discussion agent, and empirically evaluates the resulting agent.  It makes both theoretical and empirical contributions.

The reviews were almost uniformly positive. They found the theoretical analysis solid and informative, and were largely satisfied by the empirical evaluations.  Two weaknesses were commonly noted, neither of which were fatal flaws: The need to manually bucket discussion tactics and the resulting specificity to ONUW; and the lack of evaluations against human players.